# Supercoiling and looping promote DNA base accessibility and coordination among distant sites

Jonathan M. Fogg[1,2,3], Allison K. Judge [2], Erik Stricker [1], Hilda L. Chan[4,5] & Lynn Zechiedrich [1,2,3,4 ✉]

DNA in cells is supercoiled and constrained into loops and this supercoiling and looping influence every aspect of DNA activity. We show here that negative supercoiling transmits mechanical stress along the DNA backbone to disrupt base pairing at specific distant sites. Cooperativity among distant sites localizes certain sequences to superhelical apices. Base pair disruption allows sharp bending at superhelical apices, which facilitates DNA writhing to relieve torsional strain. The coupling of these processes may help prevent extensive denaturation associated with genomic instability. Our results provide a model for how DNA can form short loops, which are required for many essential processes, and how cells may use DNA loops to position nicks to facilitate repair. Furthermore, our results reveal a complex interplay between site-specific disruptions to base pairing and the 3-D conformation of DNA, which influences how genomes are stored, replicated, transcribed, repaired, and many other aspects of DNA activity.

[1] Department of Molecular Virology and Microbiology, Houston, TX, USA. [2] Verna and Marrs McLean Department of Biochemistry and Molecular Biology, Houston, TX, USA. [3] Department of Pharmacology and Chemical Biology, Houston, TX, USA. [4] Graduate Program in Immunology and Microbiology, Houston, TX, USA. [5] Medical Scientist Training Program, Baylor College of Medicine, One Baylor Plaza, Houston, TX, USA. ✉email: elz@bcm.edu

One of the properties of DNA that make it ideal as a medium for storing genetic information is its incredible stability, perhaps best illustrated by the fact that DNA samples have been recovered and sequenced from fossils over 300,000 years old[1–3]. Researchers have utilized this stability, and the rich data-density of DNA, for long-term storage of digital information[4,5]. The bases that carry the genetic code are tucked safely in the interior of double-helical B-form DNA, protecting them from damage. This hereditary information needs to be constantly accessed in living cells, thereby necessitating DNA supercoiling.

Organisms have evolved DNA supercoiling to toggle between two seemingly contradictory requirements: stable and protected DNA, and a readable genetic code. In genomes across the kingdoms of life, DNA is not only supercoiled but also constrained into loops. DNA loops, including ones with lengths as short as ~100 bp, serve important roles in gene regulation, packaging of DNA in viral capsids, and nucleosome wrapping[6–10]. Extrachromosomal loops of DNA of a wide variety of lengths, some as short as 80 bp, have been detected in the cells of various organisms, including plants, humans, mice, and C. elegans[11–13]. Supercoiling and looping both play key roles in biology. Therefore, it is important to understand how they together modulate the properties of DNA.

DNA supercoiling refers to the underwinding (negative supercoiling) or overwinding (positive supercoiling) of the DNA double helix. In most, if not all, organisms, DNA is maintained homeostatically in an underwound state[14–17]. Negative supercoiling is significantly constrained by architectural proteins. In the bacterial and eukaryotic organisms studied, however, results indicate that as much as half of negative supercoiling is unconstrained[18–27] (reviewed in refs. [14,28]). Even in the constrained portions of the genome, supercoiling strongly influences DNA structure and its interactions with proteins and other biomolecules[29,30], although supercoiling effects are even more pronounced in the unconstrained regions. DNA is not merely a passive participant in various cellular processes, but instead actively influences or even drives them[29]. The torsional strain arising from negative supercoiling reduces the energy required for strand separation. Negative supercoiling, therefore, stimulates processes that require an opening of the DNA helix such as, for example, the initiation of transcription and replication[26,31,32].

DNA supercoiling is defined by the linking number ($Lk$)[33], the total number of times the two single DNA strands coil about one another. The $Lk$ for fully relaxed B-DNA with no supercoiling, $Lk_0$, is equal to the total number of base pairs divided by the helical repeat. Thus, one measure of supercoiling is obtained by comparing the $Lk$ to $Lk_0$. This linking number difference ($\Delta Lk$), defined by $\Delta Lk = Lk - Lk_0$, is manifested through changes in twist and writhe. Twist describes the coiling of the two single strands of DNA around each other in the double helix. Writhe describes the coiling of DNA double helices about each other. How supercoiling is partitioned between twist and writhe has important biological consequences (reviewed in refs. [29,34]). Changes in twist impart helical torsional strain, much like an overwound or underwound spring. Writhing can relieve some of this torsional strain but requires the bending of stiff DNA. Furthermore, electrostatic repulsion must be overcome to bring DNA helices closer together. $\Delta Lk$ is typically scaled to the DNA length to give the superhelical density ($\sigma$) = $\Delta Lk/Lk_0$. The steady-state level of supercoiling in cells is around $\sigma = \sim -0.06$, but the superhelical density at the local level can vary considerably. Much higher levels of both negative supercoiling and positive supercoiling occur transiently[35].

DNA supercoiling levels modulate transcription to provide a mechanism of global regulation of gene expression in bacteria and eukaryotes[16,36–40]. Just as supercoiling regulates gene expression, transcription itself generates powerful transient waves of supercoiling. Positive supercoiling is generated ahead of RNA polymerase and hyper-negative supercoiling is generated behind it[41–44]. Waves of supercoiling can propagate across distances over 1 kb along the genome to drive structural transitions far away from the promoter[45–47]. These structural transitions act as a DNA "molecular servomechanism" that both senses and regulates transcription, thus providing an additional level of gene regulation[35,39,48,49]. The discovery of these long-lived supercoiling effects was surprising because it was long assumed that topoisomerases were in high enough abundance to keep pace with transcription to prevent, or at least dampen, the waves of supercoiling. These findings show that DNA supercoiling-mediated structural effects occur even in the cellular environment rich with DNA binding proteins and topoisomerases.

DNA loops found in nature are often comparable in size to, or even smaller than, the persistence length of DNA (~150 bp). DNA looping on these short-length scales requires considerable bending strain to be overcome. The need to constrain DNA into small loops such as around a nucleosome (147 bp) has led to considerable debate on how this sharp bending is achieved[50–55]. Despite the extensive and long-term effort, the question of how rigid helical DNA forms loops is still unresolved. It is generally assumed that proteins provide the energy required to overcome the intrinsic rigidity of DNA, but this conjecture fails to consider the potential contribution of DNA supercoiling. Indeed, most of the published studies on DNA flexibility employed torsionally relaxed DNA, thus neglecting any contribution from supercoiling.

We used supercoiled DNA minicircles of a few hundred base pairs to study how DNA supercoiling and looping modulate DNA structure and impact DNA activities[56–58]. When we investigated the 3D structures of supercoiled 336 bp minicircles using electron cryotomography (cryo-ET), we observed a surprisingly wide variety of minicircle shapes depending on specific supercoiling level (topoisomer)[58]. Many of the conformations observed in the cryo-ET study contained sharply bent DNA. This observation was unexpected because, as mentioned above, the DNA helix resists twisting and bending. The degree of bending we observed is not consistent with traditional models of DNA, e.g., the wormlike chain model[59–61], which largely treats the molecule as a simple isotropic elastic polymer. From these models we would expect bending and torsional strain to be distributed evenly throughout the molecule, resulting in smooth bending.

To explain our observations, we hypothesized that localized disruptions to base pairing allow the DNA to sharply bend, and thus adopt conformations that would be otherwise energetically unfavorable[58]. To test this hypothesis, here we quantified and mapped exposed bases in the 336 bp supercoiled minicircles. These new findings help to make sense of our earlier DNA simulation[62,63] and cryo-ET data[58] with these same minicircle topoisomers, and reveal an unexpected and remarkable interplay among DNA supercoiling-induced sequence-dependent disruptions to base pairing, DNA looping, and the overall shape of the DNA molecule. We also uncovered striking mechanical cooperativity in which the effects of localized disruptions at one site are communicated along the DNA backbone to cause localized helical disruptions at distant sequences. Cooperative effects between distant DNA sites have been observed experimentally[64] and in simulations[65] by studying very small DNA circles (~100 bp) unable to writhe. Our results expand our understanding of this phenomenon in two ways: first, by investigating the cooperativity in a DNA minicircle that is able to writhe and adopt a wide variety of conformations; second, by associating base pair disruptions to the cooperativity and mapping where the base pair disruptions occur. This cooperativity surely influences

the DNA shape. In turn, DNA shape surely influences exposed bases. Combined, our results offer a model for how DNA looping positions nicks and gaps for repair, how looping and supercoiling prevent genetic instability that would arise from long single-stranded stretches of DNA, and provide a mechanism for how supercoiling and looping may contribute to the three-dimensional organization of the genome. These data thus provide new molecular mechanistic insight into how DNA looping and supercoiling together regulate cellular activity.

## Results

**Study design**. The goal of this study was to understand how curvature and negative or positive supercoiling promote disruptions to DNA base pairing. To achieve this goal, we used 336 bp DNA minicircles with defined supercoiling levels for which we have extensive 3D data[58]. 336 bp minicircle DNA in its relaxed state ($Lk_0$) has 32 helical turns ($Lk = 32$; $\Delta Lk = 0$). We generated ten unique 336 bp minicircle topoisomers, ranging from hyper-negatively supercoiled ($Lk = 26$, $\Delta Lk = -6$, $\sigma = -0.189$) to positively supercoiled ($Lk = 35$, $\Delta Lk = +3$, $\sigma = +0.092$). 333, 339, 666, and 672 bp minicircles provide additional topoisomers useful for more precisely differentiating dramatic supercoil-dependent effects. Complete details for all the topoisomers studied, including $Lk$, $\Delta Lk$, and $\sigma$, as well as the calculated enzymatic cleavage rates, are listed in Supplementary Table 1.

To detect and quantify helical disruptions, we probed minicircles for exposed DNA bases with two different nucleases, Bal-31 and S1. Both enzymes cleave single-stranded DNA; however, Bal-31 also recognizes more subtle structural perturbations[66] as small as a single extrahelical base[67] or a single abasic site[68]. S1 nuclease requires a stretch of single-stranded DNA extending over at least four bases to cleave[69]. Although the structure-specific endonuclease activity of Bal-31 has well-proven utility for probing disruptions in the DNA helix[64,70–72], the precise details of what it recognizes are unknown. To use this enzyme to better understand how supercoiling exposes DNA bases, we needed to first characterize Bal-31 activity with different types of distortion and probe its molecular mechanism. With our improved understanding of Bal-31, and the inclusion of S1 nuclease to differentiate the lengths of base-pair disruption, we were able to precisely quantify and map where supercoiling-induced base-pair disruptions occurred along the DNA minicircle backbone.

**Bal-31 provides a direct and quantitative measure of exposed DNA bases**. Bal-31 cleaved the hyper-negatively supercoiled ($Lk = 26$; $\Delta Lk = -6$; $\sigma = -0.189$) 336 bp minicircle topoisomer with a concomitant rapid appearance of nicked and linear intermediates (Fig. 1a). The initial action of Bal-31 on supercoiled DNA is presumably the cleavage of a single phosphodiester bond to generate nicked DNA. The DNA is subsequently linearized by Bal-31-mediated cleavage on the strand opposite to the initial cleavage, thus invoking a nick and counter-nick mechanism.

The initial linear intermediates migrated as two distinct bands. The major band migrated anomalously slowly compared to the linear marker and ladder. Intrinsic curvature in the λ-integrase site, attR, a byproduct of the recombination process used to make the minicircles, slows migration of the linear DNA fragments on polyacrylamide gels[73]. The extent to which the intrinsic curvature affects the migration of the DNA varied when the DNA was cleaved at different restriction enzyme sites in a control experiment (Supplementary Fig. 1), supporting the idea that intrinsic curvature is causing the slow migration of the linearized minicircle. The farther the bend is from the DNA ends, the slower the migration[74]. The observed slow migration of the

linear intermediate suggests that the major Bal-31 cleavage site is located far from the site of intrinsic curvature, placing the resultant DNA ends far from the bend. The faster-migrating band (Fig. 1a) likely results from at least one additional Bal-31 cleavage site that is located much closer to the site of intrinsic curvature.

Over time, the linearized DNA became shorter, and the bands became gradually more diffuse as a result of the exonuclease activity of Bal-31, which made precise quantitation of the linear intermediates impossible. Consequently, cleavage rates were measured by quantifying the disappearance of supercoiled DNA. The disappearance of supercoiled DNA over time fitted to a linear slope (Fig. 1b). This slope did not change significantly over time, even as the substrate became depleted. From the analysis of the Michaelis-Menten kinetics of the reaction (Supplementary Fig. 2), we conclude that the $K_M$ for the endonuclease activity of Bal-31 is much higher than the DNA concentrations employed. Consequently, the Bal-31 cleavage rate was directly proportional to DNA concentration and did not plateau as the DNA concentration increased. Therefore, the enzyme does not become saturated with substrate (exposed bases), even for the most negatively supercoiled topoisomers. These results demonstrate that comparing relative Bal-31 cleavage rates among the different minicircle topoisomers provides a direct measure of the frequency of unique sites containing exposed bases.

In contrast to the rapidly degraded Bal-31 intermediates, 336 bp minicircle DNA nicked by Nb.BbvCI was a very poor substrate for Bal-31. The Nb.BbvCI-nicked species persisted for over 5 h (Fig. 2a) under conditions where negatively supercoiled DNA was degraded by Bal-31 in less than a minute (Fig. 1a). Bal-31 removes several nucleotides at the site of the initial nick[75], generating a single-stranded gap, prior to cleavage of the opposite strand. Nb.BbvCI generates a single nick at a specific DNA sequence but does not leave a gap. Gapping of the intermediate accelerates the cleavage of the strand opposite to the initial cleavage to generate linear DNA. As a control, a gapped circle was generated by removing bases next to the nick site of the Nb.BbvCI-nicked species using exonuclease III. This gapped control was cleaved extremely rapidly by Bal-31 (Supplementary Fig. 3a), supporting this explanation.

To better understand why the strand opposite to the initial nicking is cleaved at a variety of different rates (Fig. 1b), the Bal-31-nicked intermediate present after 1 min incubation was gel-purified as described in "Methods." When incubated with fresh Bal-31, the intermediate was rapidly converted to linear DNA (Supplementary Fig. 3c). The initial Bal-31 cleavage rate of the Bal-31 intermediate ($4.31 \times 10^{-3}$ s$^{-1}$; Supplementary Fig. 3d) was 180-fold faster than the cleavage rate on Nb.BbvCI-nicked DNA ($2.40 \times 10^{-5}$ s$^{-1}$; Fig. 2b). After 2 min incubation, the cleavage rate of the intermediate slowed ~10-fold ($4.07 \times 10^{-4}$ s$^{-1}$; Supplementary Fig. 3d) but was still considerably faster than the rate on DNA lacking a gap. A small fraction (13%) of the intermediate remained even after 1 h incubation. We tested whether the Bal-31-generated nick could be repaired by T4 DNA ligase. Although T4 DNA ligase is able to ligate across a gap, it does so with very low efficiency and orders of magnitude more slowly than single-nicked DNA[76]. Only 17% of the Bal-31 nicked intermediate could be repaired by T4 DNA ligase, suggesting that most of the intermediate is gapped (Supplementary Fig. 3e). The small fraction that could be repaired by T4 DNA ligase presumably lacks a gap and is, thus, resistant to cleavage by Bal-31, likely explaining why ~13% of the Bal-31 nicked intermediate remained even after 1 h incubation with Bal-31 (Supplementary Fig. 3d). When the Bal-31 intermediate was analyzed in more detail, a variety of gap sizes were observed, as evidenced by the heterogenous electrophoretic migration of the

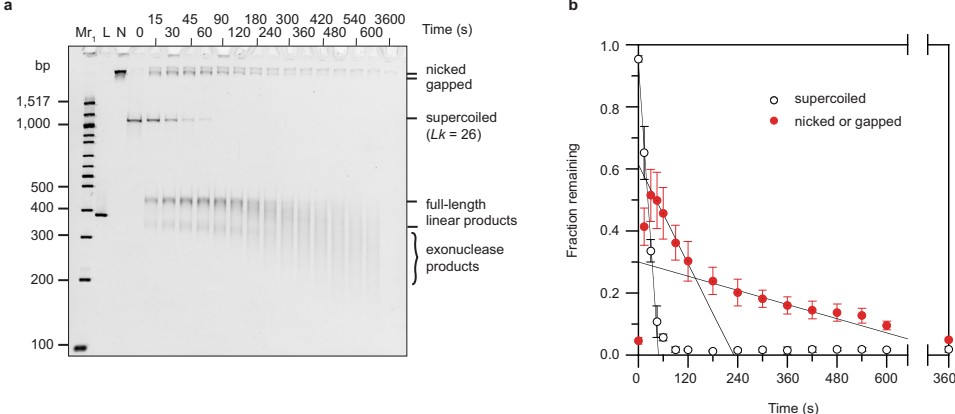

**Fig. 1 Recognition and cleavage of exposed bases in supercoiled minicircle DNA by Bal-31. a** 336 bp minicircle DNA ($Lk = 26$; $\Delta Lk = -6$; $\sigma = -0.189$) was incubated with Bal-31. At the times indicated, samples were removed, quenched by the addition of stop buffer, and analyzed by polyacrylamide gel electrophoresis. $Mr_1$: 100 bp DNA ladder; L: linear 336 bp DNA (minicircle cleaved by EcoRV); N: nicked (336 bp minicircle nicked with Nb.BbvCI). **b** The fraction of supercoiled and nicked DNA in (**a**) was quantified. The assay was repeated three times and the mean values are shown. Error bars represent standard deviations; when not visible, they were smaller than the symbol. The disappearance of supercoiled substrate over time was fit to a linear slope for the first few timepoints only. Quantifying the disappearance of the nicked (and gapped) intermediates over the extended time course revealed multiphasic degradation kinetics. The disappearance of nicked DNA over time was fit to two different linear slopes with the lines extrapolated to the axis limits to facilitate the comparison. Commercial preparations of Bal-31, as used in this assay, contain a mixture of "fast" and "slow" species of the enzyme[77]. The "fast" species generates a larger gap (~6 nucleotides) on average than the "slow" species (~3 nucleotides)[75]. The variation in gap sizes, with larger gaps, probably being cleaved more rapidly by Bal-31, likely explains the multiplicity of second-strand cleavage rates.

intermediate (Supplementary Fig. 3e). Larger gaps are presumably cleaved more rapidly by Bal-31 thus explaining the multiphasic degradation kinetics of the Bal-31 intermediate (Supplementary Fig. 3d). Bal-31 preparations typically contain a mixture of "fast" and "slow" species of the enzyme[77], which may explain the variation in gap lengths observed. It should be noted that DNA was in excess over enzyme for the steady-state kinetic assays employed in this study. Therefore, the different rates for the second-strand cleavage represent populations of DNA with different gap lengths and not different populations of enzyme.

**Constraining DNA into loops and helical phasing exposes DNA bases at a nick.** Although Bal-31 can recognize the exposed bases at a nick, cleavage is very inefficient compared to DNA with a single-stranded gap. When the crystal structure of a 12 bp nicked DNA duplex was solved by Alexander Rich and colleagues, only very limited distortions were observed at the nick site[78]. An NMR study of nicked DNA duplexes enabled a similar conclusion to be reached[79]. Base stacking interactions were proposed to be sufficient to overcome the loss of DNA strand connectivity. The lack of major distortion at the nick site raised the question of how nicks are detected by ligases and other DNA repair enzymes[78].

The base pairs on either side of the nick in short DNA duplexes, as used in the structural studies cited above, are free to adopt the most energetically favorable rotational alignment to maximize base stacking. For nicks in circular molecules, including the minicircles used in the current study, the situation is more constrained. The rotational alignment of the base pairs flanking the nick site in a circular molecule is limited; only in a loop comprising an integer number of helical turns would the base pairs flanking the nick site be aligned, which is a seemingly unlikely scenario in the cellular context. DNA in cells is also often constrained into loops, resulting in it behaving as if it is circular.

The number of helical turns of a 336 bp minicircle should be very close to a perfect integer value, placing the base pairs flanking the nick site in close rotational alignment. In contrast, the number of helical turns in the nicked 333 bp and nicked 339 bp minicircles should significantly deviate from a perfect

integer value, potentially exposing the strand ends at the nick site. That these latter minicircles were out of phase is supported by the observation that nicked 333 bp and nicked 339 bp minicircles were significantly better substrates for Bal-31 than nicked 336 bp minicircles (Fig. 2a); nicked 333 bp minicircle was cleaved ~three-fold more rapidly ($6.23 \times 10^{-5}\,s^{-1}$; Fig. 2b); nicked 339 bp minicircle was cleaved ~six-fold more rapidly ($1.41 \times 10^{-4}\,s^{-1}$; Fig. 2b).

Differences in phasing can be observed by comparing the electrophoretic mobility of minicircle topoisomers with $Lk$ on either side ($\Delta Lk = \pm 1$) of the $Lk = 32$ topoisomer (Fig. 2c). This method of analysis provides a means to determine the $Lk_0$ and, thus, to directly calculate the helical repeat of DNA[80], which is sensitive to solution conditions[81,82]. We performed the analysis under buffer conditions matching those used for the Bal-31 cleavage experiments (600 mM NaCl, 12 mM $CaCl_2$, 12 mM $MgCl_2$). For the 336 bp minicircle, the $Lk = 31$; $\Delta Lk = -1$ and $Lk = 33$; $\Delta Lk = +1$ topoisomers had very similar mobility, indicating that these topoisomers were equally different from the $Lk_0$. Therefore, the $Lk_0$ must be very close to 32 under these buffer conditions. In contrast, the mobilities of the $Lk = 31$; $\Delta Lk = -1$ and $Lk = 33$; $\Delta Lk = +1$ topoisomers of the 333 bp minicircle differed from each other with the $Lk = 33$ migrating further. Faster migration of the $Lk = 33$ topoisomer indicates that the $Lk_0$ for the 333 bp minicircle must be less than 32. For the 339 bp minicircle, the difference between the $Lk = 31$; $\Delta Lk = -1$ and $Lk = 33$; $\Delta Lk = +1$ topoisomers was even more pronounced but this time the $Lk = 31$ migrated further. From the comparison of all these relative electrophoretic mobilities (see "Methods"), we calculated the helical repeat, under Bal-31 reaction conditions, to be ~10.48 bp/turn. The calculations of superhelical density listed in Supplementary Table 1 were determined using this helical repeat value.

A helical repeat value of 10.48 bp/turn translates to $Lk_0$ values of 32.06 for the 336 bp minicircle, 31.78 for the 333 bp minicircle, and 32.35 for the 339 bp minicircle under these buffer conditions. The nicked 333 bp minicircle is approximately one-fifth of a turn out-of-phase (Fig. 2a), making it more susceptible to Bal-31. The nicked 339 bp minicircle is the most out-of-phase (approximately

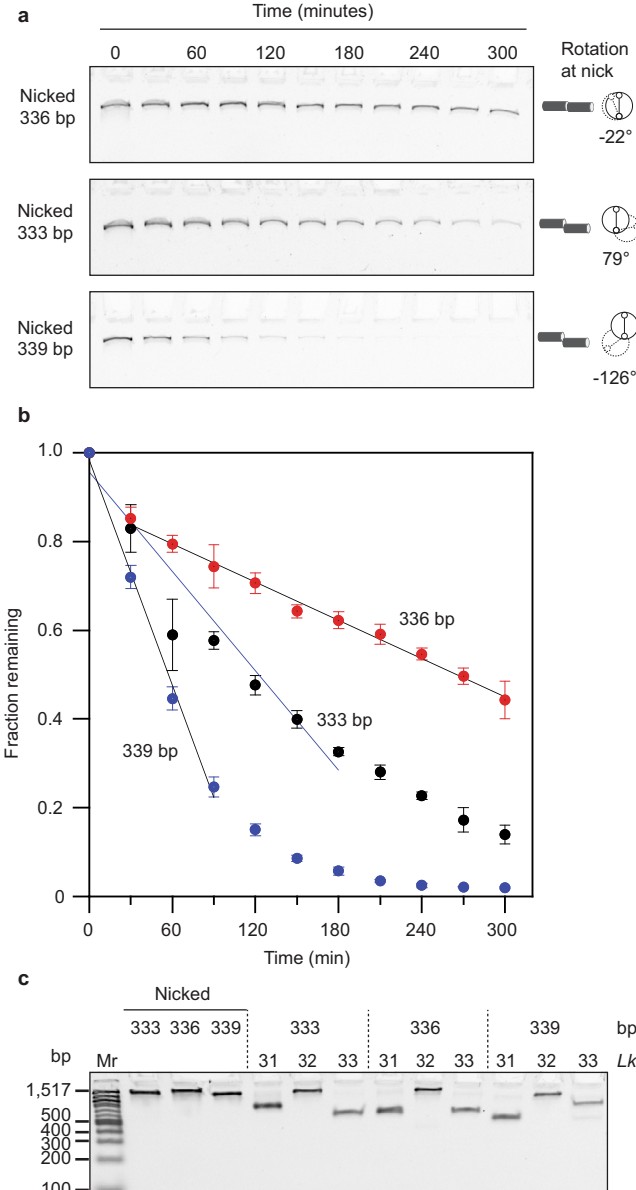

**Fig. 2 Base exposure in nicked DNA depends upon DNA phasing. a** 336, 333, or 339 bp minicircle DNA, previously nicked by Nb.BbvCI, was incubated with Bal-31. At the times indicated, samples were removed, quenched by the addition of stop buffer, and analyzed by polyacrylamide gel electrophoresis. Diagrams to the right of each gel show the predicted rotational alignment of the DNA ends on either side of the nick site. Uncropped gel images are provided in Source Data. **b** The fraction of nicked DNA remaining over time in (**a**) was quantified. The assay was repeated three times and mean values are shown. Error bars represent standard deviations. The data were fitted to a linear slope, excluding the later timepoints for the 333 bp and 339 bp minicircles that deviated from the linear relationship. **c** Relative electrophoretic mobility of minicircle topoisomers to determine $Lk_O$. The polyacrylamide gel was run under conditions replicating Bal-31 reaction conditions (600 mM NaCl, 12 mM CaCl₂, 12 mM MgCl₂). This assay was performed one time.

one-third of a turn; Fig. 2a), and consequently is most susceptible to Bal-31. Interestingly, the out-of-phase nicked 333 and 339 bp minicircles migrate faster than the in-phase nicked 336 bp minicircle (Fig. 2c), which may be explained by kinking at the nick site because of reduced base stacking. Disruption of base stacking, through the addition of urea, at nick sites in DNA

duplexes has been previously shown to induce a kink in the DNA[83,84]. A kink at the nick site in the out-of-phase minicircles may induce the DNA to adopt a more compact teardrop or elliptical shape, resulting in faster migration on the gel. These data reveal an important consequence of constraining DNA into loops. DNA looping may position nicks to be recognized by ligase and other DNA repair enzymes.

**Changes in negative supercoiling modulate the location of DNA base exposure.** A DNA conformation closely associated with DNA supercoiling is the plectonemic (interwound) super-helix, which is the consequence of writhe. Writhing requires sharp bending of the DNA helix especially at superhelical apices, the loops found at the ends of the plectonemes. The degree of bending at superhelical apices is inversely related to the size of the apical loop; the smaller the loop, the more pronounced the bending. We previously identified a hotspot for Bal-31 cleavage in the most highly negatively supercoiled ($Lk = 26$; $\Delta Lk = -6$; $\sigma = -0.189$) 336 bp minicircle topoisomer[58]. This topoisomer adopts highly writhed conformations, primarily rod-like shapes with very small loops at the superhelical apices. In comparison to double-stranded DNA, single-stranded DNA is very flexible[85]; and therefore, supercoiling-induced base pair disruptions may provide a hyperflexible hinge to facilitate this bending[86–89]. Conversely, sharp bending, and the concomitant bending strain, directly disrupts base pairing[70,90–92]. We speculated that the denaturation site recognized by Bal-31 localizes to one of the two apices of rod-shaped minicircles where the DNA is sharply bent[58], suggesting a relationship between denaturation and DNA shape.

The other, less negatively supercoiled minicircle topoisomers, adopt different shapes, leading us to reason that exposed bases may occur at different locations in these topoisomers. The slow cleavage rate of Bal-31 and S1 nuclease with positively supercoiled DNA, together with the exonuclease activity of Bal-31, precluded our efforts to map cleavage sites on the positively supercoiled topoisomers. Bal-31 cleavage sites on negatively supercoiled topoisomers were mapped by isolating Bal-31-linearized DNA and subsequently cleaving this DNA with various restriction enzymes that provide sequence reference points (Fig. 3a). The resulting fragment lengths were determined by high-resolution agarose gel electrophoresis to deduce where Bal-31 cleaved relative to the known restriction sites, allowing the cleavage sites to be mapped. In contrast to polyacrylamide gels, the mobilities on agarose gels are not significantly affected by the intrinsic curvature in the minicircle (Supplementary Fig. 1). Therefore, agarose gel electrophoresis provides measurements of DNA lengths, accurate to within ~10 bp. The previously determined hotspot for Bal-31 cleavage (site 1) was found in all the negatively supercoiled topoisomers. Cleavage at this site produced a distinct pattern of fragment lengths, most evident for the $Lk = 29$; $\Delta Lk = -3$; $\sigma = -0.095$ topoisomer (Fig. 3b). By averaging the results from each of the different restriction enzymes, we mapped site 1 to sequence position 143 (±8 bp) for the $Lk = 29$ topoisomer. Mapping of S1 nuclease cleavage produced an almost identical pattern of fragments (Fig. 3c), demonstrating that S1 nuclease also cleaves the $Lk = 29$ topoisomer preferentially at site 1. Remarkably, then, we have converted both of these non-specific nucleases into site-specific enzymes by supercoiling and curvature.

For other topoisomers, we observed additional fragment lengths generated by either Bal-31 (Supplementary Fig. 4) or S1 nuclease (Supplementary Fig. 5), indicating two more hotspots (sites 2 and 3) for nuclease cleavage (Fig. 3a). The sequence positions of sites 1, 2, and 3 reported here are the average across

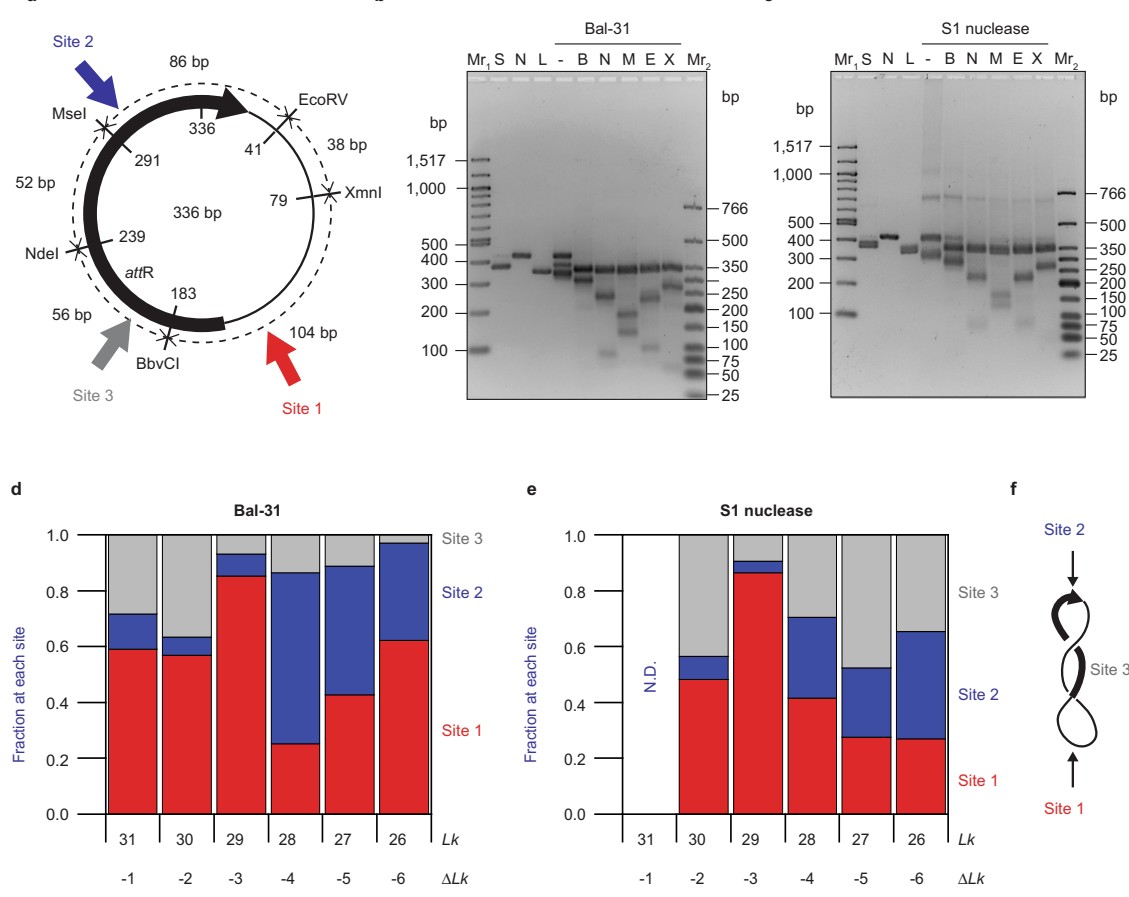

**Fig. 3 Bal-31 and S1 nuclease cleave at specific DNA sites containing exposed bases. a** Map of the 336 bp minicircle sequence showing the positions of the restriction enzymes used, the location of the three major Bal-31/S1 nuclease cleavage sites, and the *att*R integrase site. The numbers on the inside of the circle indicate the designated start/end of the minicircle (see "Methods") and also the positions of each restriction site along the listed sequence. Dashed lines indicate the distances between each restriction cleavage site. Intrinsic curvature in *att*R is centered around the MseI site (Supplementary Fig. 1). **b** Minicircle DNA was cleaved with Bal-31, deproteinized, then subsequently cleaved with each of the restriction enzymes, and the products were separated by agarose gel electrophoresis. The 336 bp $Lk = 29$; $\Delta Lk = -3$; $\sigma = -0.095$ topoisomer is shown. $Mr_1$: 100 bp DNA ladder; $Mr_2$: low molecular weight DNA ladder; S: supercoiled (336 bp $Lk = 29$; $\Delta Lk = -3$), N: nicked 336 bp minicircle; L: 336 bp minicircle linearized by EcoRV; -, B, N, M, E, X: minicircle incubated with Bal-31 for 1 min followed by incubation with a second restriction enzyme as indicated (-: no second enzyme; B: BbvCI; N: NdeI; M: MseI; E: EcoRV; X: XmnI). This assay was performed at least two times for each topoisomer with very similar results. **c** Mapping of S1 nuclease cleavage sites (336 bp; $Lk = 29$; $\Delta Lk = -3$ topoisomer is shown). The experiment was performed following the same protocol as for Bal-31. This assay was performed once for each topoisomer. **d** Relative Bal-31 cleavage at each of the three sites as a function of $Lk$. **e** Relative S1 nuclease cleavage at each of the three sites as a function of $Lk$. N.D. not determined. **f** Model showing localization of sites 1 and 2 to the superhelical apices.

all topoisomers for Bal-31. Mapping results for each individual topoisomer are listed for Bal-31 (Supplementary Table 2) and S1 nuclease (Supplementary Table 3). For each site, Bal-31 and S1 nuclease cleavage mapped to within a 20 bp region. The presence of gaps of varying lengths in the Bal-31 intermediate, and multiple cleavage sites in the larger denaturation bubbles cleaved by S1 nuclease, made it impossible to resolve cleavage to the nucleotide level. Quantitation of the nuclease cleavage site preference (Supplementary Fig. 6) revealed clear differences among the different topoisomers for both Bal-31 (Fig. 3d) and S1 nuclease (Fig. 3e), as well as subtle but instructive differences between the two enzymes. Once the DNA is initially nicked by either Bal-31 or S1 nuclease, the torsional strain will be released leading to a loss of the nuclease susceptible site. Thus, Bal-31 and S1 nuclease should act only once on each minicircle. The Bal-31 cleavage preferences reflect the relative frequency of exposed bases at each of the three sites whereas S1 nuclease preferences reflect the relative frequency of longer (>4 nucleotide) single-stranded bubbles.

Both Bal-31 and S1 nuclease cleaved at sequence position ~302 (site 2), diametrically opposite on the minicircle to site 1, in the region predicted using the WebSIDD algorithm[93] to be the most prone to denaturation based on DNA sequence. We previously speculated that denaturation at site 2, resulting in a small single-stranded bubble with concomitant increased flexibility, may localize site 2 to an apex, which then induces base exposure at the site opposite (site 1)[58]. Base pair disruption at site 2 governs the positioning of the superhelical apices; therefore, we expected site 2 to be a major Bal-31 and S1 nuclease cleavage site. Our results, however, showed that site 2 is only a preferred Bal-31 (Fig. 3d) and S1 nuclease (Fig. 3e) cleavage site in the more negatively supercoiled topoisomers ($Lk = 28$ to 26; $\Delta Lk = -4$ to $-6$; $\sigma = -0.127$ to $-0.189$). The strong preference for site 1 over site 2 across other supercoiling values would seem to bring into question the significance of base-pair disruption at site 2. One possible molecular explanation for this unexpected result is that denaturation at site 1 may be more extensive than at site 2. It is reasonable to assume that, because Bal-31 cleaves different sized

gaps with varying efficiency (Supplementary Fig. 3), it may also cleave different sized distortions in supercoiled DNA at different rates. A single base-pair opening at site 2 may increase the flexibility of the DNA at the point where the helix is disrupted, and this increased flexibility may be enough to localize site 2 to an apex. The resulting sharp bending induced at site 1 opposite may require more extensive denaturation than at site 2, thus resulting in site 1 being the preferred site for Bal-31 cleavage.

The Bal-31 and S1 nuclease cleavage site at sequence position ~203 (site 3) differed from sites 1 and 2 in that no significant Bal-31 or S1 nuclease cleavage site was detected diametrically opposite. We hypothesize that sites 1 and 2 are each localized to one of the superhelical apices, whereas site 3 is not (Fig. 3f). Alternatively, site 3 may be localized to an apex of an asymmetric three-lobed conformation, together with sites 1 and 2; however, we did not observe shapes matching this description in our previous cryo-ET study[58]. Bal-31 cleaved site 3 in the $Lk = 31$; $\Delta Lk = -1$; $\sigma = -0.033$ and $Lk = 30$; $\Delta Lk = -2$; $\sigma = -0.064$ topoisomers; however, Bal-31 cleavage at site 3 was barely detectable in the more negatively supercoiled topoisomers (Fig. 3d). In contrast, site 3 was a preferred S1 nuclease cleavage site for all the negatively supercoiled minicircles tested, except for the $Lk = 29$; $\Delta Lk = -3$; $\sigma = -0.095$ topoisomer (Fig. 3e). For example, for the $Lk = 26$ ($\Delta Lk = -6$; $\sigma = -0.189$) topoisomer, Bal-31 cleavage at site 3 accounted for only 2% of total cleavage whereas 40% of S1 nuclease cleavage of this topoisomer was at site 3. Therefore, site 3 is not frequently denatured in the more negatively supercoiled topoisomers, but when it is, a single-stranded bubble of at least four unpaired bases is likely to form, as detected by S1 nuclease.

**DNA base exposure sharply increases at a distinct threshold of negative supercoiling**. Comparison of the Bal-31 steady-state cleavage rates (listed in Supplementary Table 1) as a function of supercoiling revealed a sharp threshold of negative supercoiling associated with a dramatic increase in Bal-31 sensitivity (Fig. 4a). Beyond this threshold, the Bal-31 cleavage rate plateaued at a high rate. Bal-31 cleaved less negatively supercoiled topoisomers at a much slower rate. No activity, indicating no base exposure, was detected until the negative supercoiling level reached $\sigma = -0.024$. With increasing negative supercoiling from $\sigma = -0.024$ to $-0.042$, there was a gradual increase in the Bal-31 cleavage rate and a more pronounced (10-fold) increase in the cleavage rate between $\sigma = -0.042$ and $-0.056$. We previously derived the probabilities of having at least one base pair opening in the 336 bp minicircle using coarse-grained simulations[63]. These probabilities were 0% for $Lk = 32$ ($\Delta Lk = 0$; $\sigma = -0.002$), 10% for $Lk = 31$ ($\Delta Lk = -1$; $\sigma = -0.033$), and ~100% for $Lk = 30$ to 26 ($\Delta Lk = -2$ to $-6$; $\sigma = -0.064$ to $-0.189$). The differences in Bal-31 cleavage rates we measured here qualitatively agree with these predicted probabilities.

A similar kinetic analysis of the supercoiling dependence of S1 nuclease activity was performed (Fig. 4b; Supplementary Table 1). The 336 bp; $Lk = 31$; $\Delta Lk = -1$; $\sigma = -0.033$ topoisomer was a very poor substrate for S1 nuclease, requiring very long incubation times (several hours). Distortions in this topoisomer are, therefore, mostly limited to less than four consecutive base pairs being disrupted. In contrast, the $Lk = 30$; $\Delta Lk = -2$; $\sigma = -0.064$ topoisomer was sensitive to S1 nuclease, indicating more extensive denaturation. The cleavage rate for S1 nuclease further increased ~14-fold between $\sigma = -0.064$ and $-0.0189$ (Fig. 4b), whereas the rate for Bal-31 did not significantly change over this range (Fig. 4a). The plateau in Bal-31 cleavage rates suggests that the frequency of unique Bal-31 cleavage sites does not significantly change between $\sigma = -0.064$ and $-0.0189$.

Instead, the length of the single-stranded bubbles increases, as evidenced by the increased sensitivity to S1 nuclease.

The sharp increase in Bal-31 activity at the corresponding threshold of superhelical density (between $\sigma = -0.042$ and $-0.056$) is presumably caused by a dramatic increase in the frequency of sites containing exposed bases. At lower levels of negative supercoiling, base-pair disruption is infrequent, but beyond the threshold, base exposure occurs at a high frequency. The difference in Bal-31 cleavage rates for different sized distortions, as inferred from the different cleavage rates for different sized gaps (Supplementary Fig. 3), suggests that the increase in the frequency of sites is not the only contributing factor to the increased rate. Although Bal-31 recognizes a single exposed base in an otherwise intact helix, these minimal distortions are cleaved relatively slowly by Bal-31. The 336 bp $Lk = 30$; $\Delta Lk = -2$; $\sigma = -0.064$ topoisomer is cleaved by S1 nuclease, albeit relatively slowly, but the $Lk = 31$; $\Delta Lk = -1$; $\sigma = -0.033$ topoisomer is not (Fig. 4b), providing evidence that this threshold of maximal Bal-31 activity also coincides with the transition from minimal base pair disruptions to small single-stranded bubbles. These small single-stranded bubbles are better substrates for Bal-31 than a single disrupted base pair, contributing to the increased cleavage rate. Small single-stranded bubbles of a few unpaired bases are at the lower limit of what S1 nuclease can recognize and are cleaved relatively slowly. Further increases in the length of these denaturation sites with negative supercoiling increase the S1 cleavage rate. In contrast, the Bal-31 cleavage rate plateaus beyond the threshold, suggesting that a small bubble consisting of a few unpaired bases is sufficient for maximal Bal-31 cleavage activity. Although these bubbles continue to increase in length with increasing negative super-coiling, this additional increase in bubble size does not further increase the Bal-31 cleavage rate.

**Positive supercoiling exposes DNA bases**. Positive supercoiling is generated ahead of replication and transcription machinery as they progress along the DNA template[42–44]. Nucleosomes in chromatin limit the intertwining of the newly replicated daughter duplexes behind the polymerase during replication, driving the supercoiling to partition primarily ahead of the replication fork[94]. A similar nucleosome-driven preferential partitioning of super-coiling into positive supercoiling ahead of the polymerase may also apply to transcription. Unless this positive supercoiling is efficiently relaxed, the polymerases will stall, and replication and transcription will grind to a halt. Some topoisomerases pre-ferentially relax positively supercoiled DNA to facilitate these processes (although how these enzymes differentiate positive and negative supercoiling is not fully understood)[34,95,96].

A prime motivation to study positively supercoiled DNA is that important antibacterial drugs (e.g., fluoroquinolones) and anti-cancer drugs (e.g., epipodophyllotoxins, anthracyclines, and mitoxantrone) target the topoisomerases that preferentially act on positively supercoiled DNA[97–99]. Despite the importance of positively supercoiled DNA, it has not been studied to the same extent as negatively supercoiled DNA.

Overwinding of the helix should make DNA resistant to strand separation, but we detected Bal-31 cleavage, indicative of exposed bases, in positively supercoiled DNA. The threshold for Bal-31 activity was much higher for positively supercoiled than for negatively supercoiled DNA (Fig. 4a). No Bal-31 cleavage was detected until $\sigma = +0.060$ (the equivalent threshold for negatively supercoiled DNA falls between $\sigma = -0.011$ and $\sigma = -0.024$ (Fig. 4a)). Beyond this threshold, Bal-31 sensitivity increased steadily with increasing positive supercoiling (Fig. 4a). The maximal cleavage rate ($2.27 \times 10^{-4} \text{ s}^{-1}$) observed with positively

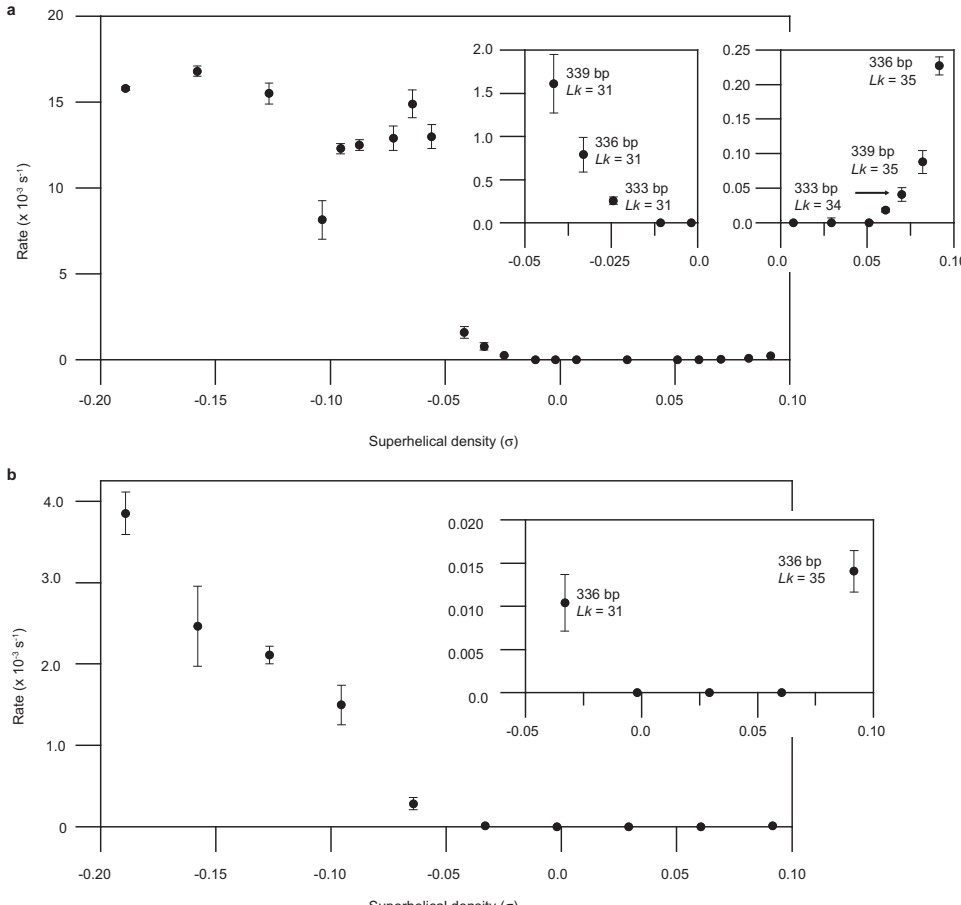

**Fig. 4 DNA base exposure increases at a sharp threshold of supercoiling. a** Comparison of Bal-31 cleavage rates as a function of superhelical density. Left inset, rescaled *y*-axis for data range $\sigma = 0$ to $\sigma = -0.05$ to show the onset of Bal-31 activity with negative supercoiling. Right inset, rescaled *y*-axis for data range $\sigma = 0$ to $\sigma = +0.10$ to show Bal-31 activity on positively supercoiled DNA. **b** Comparison of S1 nuclease cleavage rates as a function of superhelical density. Inset, rescaled *y*-axis for data range $\sigma = -0.05$ to $\sigma = +0.10$. Cleavage rates for Bal-31 and S1 nuclease were determined as described in "Methods". Each kinetic assay was repeated at least three times and the mean values are shown. Error bars represent standard deviations.

supercoiled (336 bp; $Lk = 35$; $\Delta Lk = +3$) DNA, however, was 74-fold slower than the maximal rate ($1.68 \times 10^{-2}\,\text{s}^{-1}$) observed with negatively supercoiled DNA (Supplementary Table 1). Our previously published coarse-grained simulations predicted a relatively low frequency (4.5%) of base-pair opening in the $Lk = 35$ topoisomer[63], compared to ~100% in the equivalent negatively supercoiled topoisomer ($Lk = 29$), perhaps at least partially explaining these differences in Bal-31 cleavage rates.

The 336 bp; $Lk = 35$; $\Delta Lk = +3$; $\sigma = +0.092$ topoisomer was the only positively supercoiled topoisomer tested for which there was detectable S1 nuclease activity (Fig. 4b). The S1 nuclease-mediated cleavage of this topoisomer was extremely slow, requiring several hours of incubation for the significant activity to be detected. The paucity of S1-susceptible sites in this topoisomer indicates that the disruptions are limited to less than four consecutive base pairs. The slow Bal-31 cleavage rate and absence of significant S1 nuclease activity together suggest that although exposed bases are found in positively supercoiled DNA, they occur infrequently, and sites are limited in length.

**DNA looping facilitates base exposure and modulates its location.** As mentioned above, we previously proposed a model whereby denaturation promotes sharp bending and, conversely, bending stress promotes localized denaturation[58]. The increased mechanical stress that arises from constraining DNA into a small loop should increase denaturation. As the loop length gets

shorter, and especially as it approaches the persistence length of DNA, bending strain increases. To test this reasoning, we performed a kinetic analysis of Bal-31 cleavage of double-length (666 and 672 bp) minicircles that should be less susceptible to denaturation than the twice shorter minicircles.

Similar to the 333, 336, and 339 bp minicircles, the 666 and 672 bp topoisomers with negative supercoiling beyond a certain threshold were cleaved very rapidly by Bal-31 (Fig. 5a). Whereas for 333, 336, and 339 bp minicircles this threshold occurred at σ ~ −0.05, for the twice larger minicircles this threshold shifted to σ ~ −0.07 (Fig. 5b). We conclude that increased curvature in the smaller minicircles makes denaturation more favorable and therefore less supercoiling is required for maximal Bal-31 cleavage. This finding suggests the intriguing possibility that varying DNA loop lengths may provide the cell yet another layer of control of DNA activities, including, for example, transcription.

The double-length minicircles have two copies of the sequence of the 333 or 336 bp minicircle in tandem orientation. When we mapped the Bal-31 cleavage site in the 672 bp $Lk = 58$; $\Delta Lk = -6$; $\sigma = -0.096$ minicircle topoisomer, we only observed cleavage at site 2 (Fig. 5c). In comparison, the equivalent topoisomer (336 bp; $Lk = 29$; $\Delta Lk = -3$; $\sigma = -0.095$) in the single-length minicircles was cleaved almost exclusively at site 1. One interpretation of these data is that site 2 preferentially localizes to one of the superhelical apices. In the 336 bp minicircle, the site opposite site

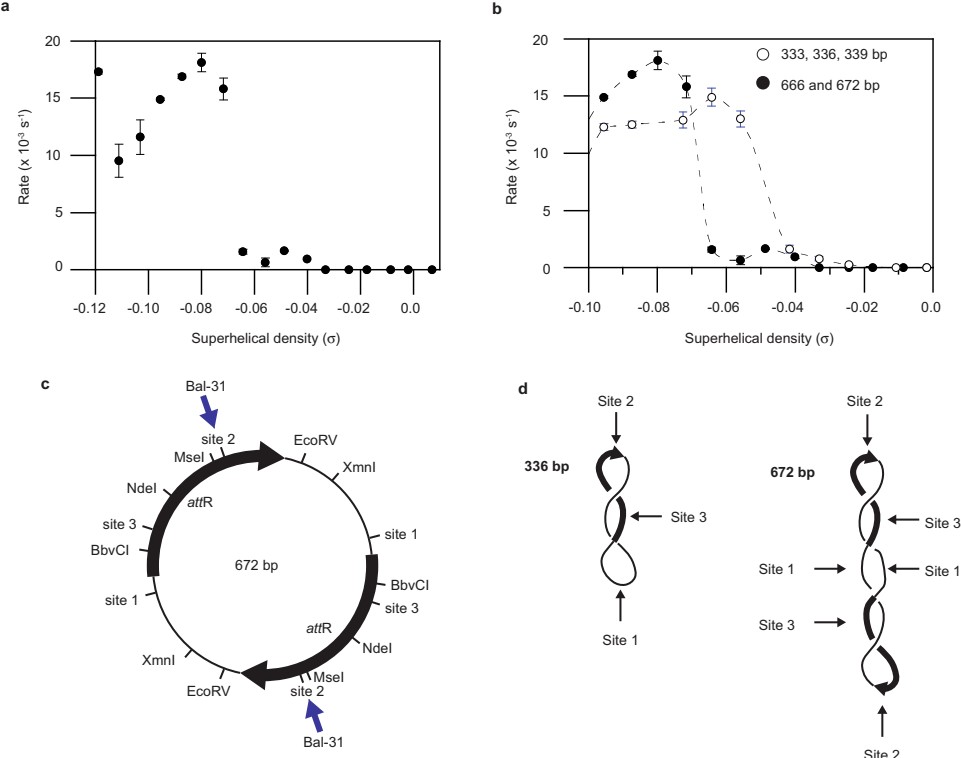

**Fig. 5 Curvature and sequence affect supercoiling threshold for DNA base exposure. a** Bal-31 cleavage rates of double-sized (666 and 672 bp) minicircle topoisomers as a function of superhelical density. The assay was repeated three times for each topoisomer and mean values are shown. Error bars represent standard deviations; when not visible, they were smaller than the symbol. **b** Comparison of thresholds of Bal-31 activity for different sized minicircles. Data from Figs. 4a and 5a were overlaid on the same plot to facilitate comparison. Fitted lines are for visualization purposes only. **c** Map of the double-length minicircle sequence. Arrows indicate the position of Bal-31 cleavage of the 672 bp $Lk = 58$ ($\Delta Lk = -6$; $\sigma = -0.096$) topoisomer (see Source Data for these results). Double-length minicircles employed in this study have two copies of the monomeric minicircle sequence in direct orientation. **d** Model for why Bal-31 preferentially cleaves site 1 in the 336 bp minicircle but site 2 in the 672 bp minicircle. Site 2 is localized to one of the superhelical apices, placing site 1 at the opposite apex. In the 336 bp minicircle; however, in the 672 bp minicircle, another copy of site 2 is found at the opposite apex.

2 is site 1, resulting in denaturation at a site otherwise unexpected to denature (Fig. 5d). In the 672 bp minicircle, the site opposite is another copy of site 2. Bal-31 site preference is, therefore, modulated by the DNA sequence at the site opposite. We previously observed that the 672 bp minicircle predominantly adopts highly writhed and elongated conformations[58]. The cooperative effect of having two bend sites diametrically opposite each other likely accounts for the highly writhed conformation. These results revealed cooperative effects among distant DNA sites. Our findings also show how this cooperativity dictates DNA shape, providing a mechanism for regulating cellular activity.

## Discussion

We have shown that the wide variety of conformations that minicircle DNA adopts as a function of DNA supercoiling is accompanied by site-specific disruptions on the base-pair level. Localized disruptions to base-pairing—from a single base flipping to more extensive denaturation—allow short stretches of DNA to deviate from the simple elastic (all B-DNA) regime in which the torsional and bending strains are evenly distributed. Supercoiling may thus be able to stimulate biological activity by allowing DNA to adopt a multiplicity of conformations, including ones that would be energetically unfavorable if not for the disruptions to base pairing. The structural diversity of supercoiled DNA increases the possibility of accessing a conformation favorable for the binding of proteins, DNA, RNA, and other biomolecules.

Although we did not explicitly set out to determine the molecular mechanisms of Bal-31 and S1 nuclease, it was

important to understand how different types of structural perturbation are recognized and processed by these enzymes. Bal-31 recognizes a broad array of distortions; however, we found that the efficiency with which it processes them varies widely. In addition to a dramatic increase in the frequency of sites containing exposed bases, the threshold of supercoiling that led to maximal Bal-31 activity also coincided with the transition from minimal distortions to small single-stranded bubbles.

Bal-31 was drastically slower at cleaving positively supercoiled than negatively supercoiled minicircles and required a higher threshold of positive supercoiling for activity. The existence of exposed DNA bases in positively supercoiled DNA is surprising. Molecular dynamics simulations of very small (65 bp) minicircles have shown that bending strain alone, a consequence of the small loop size, is sufficient to disrupt base pairs, even in torsionally relaxed or positively supercoiled ($\sigma = +0.18$) DNA[92]. The distortion observed in the simulations of relaxed and positively supercoiled 65 bp minicircles was equivalent to the type II kinks described previously[100], consisting of a sharp bend distributed over three base pairs with a single base pair disrupted at the center of the bend. Sharp bending is a prominent feature of the highly writhed conformations we observed for the 336 bp $Lk = 35$; $\Delta Lk = +3$; $\sigma = +0.092$ topoisomer[58]. If sharp bending disrupts base pairing, as observed in the simulations, this would explain the sensitivity of this topoisomer to Bal-31.

An intriguing alternative explanation for exposed bases is that the torsional strain in these highly positively supercoiled minicircles may be sufficient to trigger the formation of P-DNA—an

inside-out DNA conformation with bases exposed on the outside of the helix and the backbone on the inside. P-DNA formation was postulated to account for observations of exposed bases in over-wound DNA held under tension to prevent the release of torsional strain through writhing[101,102]. We directly observed P-DNA in our molecular dynamics simulations of overwound DNA molecules not under tension but unable to writhe because of the periodic boundary conditions employed in the simulations[62]. The superhelical density of the $Lk = 35$ topoisomer ($\sigma = +0.092$) is much higher than the threshold value ($\sigma = +0.037$) at which P-DNA was observed in single-molecule experiments[101]. The helical repeat of P-DNA is estimated to be ~3 bp/turn[62,101,103] compared to ~10.5 bp/turn for B-DNA; therefore, conversion of ~12 bp of B-DNA to P-DNA, if it were to occur, should be sufficient to accommodate all of the overwinding in the $Lk = 35$; $\Delta Lk = +3$; $\sigma = +0.092$ topoisomer.

In general, DNA tends to become more compact and writhed with increasing negative supercoiling[58]. There was a striking deviation from this tendency in the distribution of 3D minicircle conforma-tions observed by cryo-ET[58]. The more negatively supercoiled minicircle topoisomer (336 bp; $Lk = 30$; $\Delta Lk = -2$; $\sigma = -0.064$) was observed more often in an open circular conformation than the less supercoiled ($Lk = 31$; $\Delta Lk = -1$; $\sigma = -0.033$) topoisomer[58]. Note that differences in buffer conditions, 10 mM $CaCl_2$ for cryo-ET vs. 12 mM $CaCl_2$, 12 mM $MgCl_2$, 600 mM NaCl in the current study, slightly change the helical repeat. Therefore, the superhelical densities reported in our earlier cryo-ET study slightly differ from those reported here. Our results here provide a potential explanation for this previously puzzling observation. The threshold ($\sigma = -0.056$) for maximal Bal-31 activity coincides with the negative supercoiling level associated with this shift towards more open conformations. Dena-turation ($\geq 4$ bp) in the $Lk = 30$; $\Delta Lk = -2$; $\sigma = -0.064$ topoisomer releases torsional strain, reducing the need to writhe, and thus allowing this topoisomer to adopt more open conformations. The $Lk = 31$; $\Delta Lk = -1$; $\sigma = -0.033$ topoisomer, however, does not as extensively denature and may only have disruptions as short as a single base pair, as shown by the absence of S1 nuclease activity. Therefore, more of the supercoiling must be manifested as writhe to release torsional strain. Localized disruptions to base-pairing can modulate the partitioning between twist and writhe by allowing the DNA to fluctuate between open and writhed conformations. One caveat to this discussion is that although we took several steps to ensure the purity of each minicircle topoisomer in the cryo-ET study, contamination of just the $Lk = 30$; $Lk = -2$; $\sigma = -0.064$ topoisomer with nicked minicircle (which adopts all open conformations), although highly unlikely, cannot be formally ruled out.

The three-dimensional arrangement of both bacterial and eukaryotic genomes is highly organized[104–107]. Our data sup-port the idea that specific sequences are localized to superhelical apices, which has been described as a "hidden code" that con-tributes to the spatial ordering of the genome[108]. Intrinsically curved sequences upstream of promoters may localize tran-scription start sites close to superhelical apices[108]. The place-ment of the transcription start site close to the apical loop may allow supercoiling to modulate gene expression by bringing enhancer and promoter sequences close together[109,110] or by helping RNA polymerase to initiate transcription through the formation of an open complex[40].

A recently published high-resolution atomic force microscopy and molecular dynamics study using supercoiled 339 bp mini-circles provides direct evidence of base-pair disruption at the superhelical apices and sharply bent regions[111]. Furthermore, in numerous computational simulations of supercoiled DNA, base pair disruption has been identified at the superhelical apices where the DNA is typically most sharply bent[58,65,92,100,112]. One way to accommodate sharp bending is by kinking of the DNA, which may include base pair disruptions[92,100]. Kinks observed in

computational simulations of small DNA circles[94] bend the DNA towards the minor groove, thus, relating kink formation to the rotational register of the DNA helix. Once a kink has formed at one superhelical apex, helical phasing will modulate where a kink may form at the opposite apex[92,100], such that the minor groove is preferably located on the inside of the circle, influencing the location of secondary Bal-31 cleavage sites. Other types of dis-tortion, such as DNA base-pair mismatches, also localize to superhelical apices, which may facilitate recognition and repair of these lesions[113].

Disruptions to base pairing, even as small as a single flipped base, may increase DNA flexibility, localizing a particular sequence (e.g., site 2) to a superhelical apex (Fig. 6a). Increased bending that results at the opposite superhelical apex (site 1) may lead to more extensive disruptions to base-pairing because of the bending strain, allowing the apical loops with base-pair disrup-tions to become smaller[112] and writhe to be accommodated by a smaller portion of the minicircle. This accommodation would facilitate additional writhing, which may further stimulate base-pair disruption at the first site because of increased bending strain. Thus, site 2 stimulates base-pair disruption at site 1 and, in turn, site 1 stimulates denaturation at site 2 as the DNA becomes more highly writhed. This scenario may explain why site 2 is only a preferred Bal-31 and S1 nuclease cleavage site in the more negatively supercoiled topoisomers. The cooperativity between the two sites helps to lock these sequences at the apices.

The apical localization of site 2 may also be influenced by the intrinsic curvature that is centered around this region of the *attR* site[73,114,115] (Supplementary Fig. 1). Intrinsically bent DNA preferentially localizes to superhelical apices[116], and this locali-zation enhances the efficiency of λ-integrase mediated site-specific recombination[115]. Although it is clear that DNA sequence strongly influences the positioning of superhelical apices[108], the underlying reasons have been disputed. The sequence-dependent flexibility of A-tracts may be more important in influencing shape than intrinsic curvature[117]. Indeed, the degree of bending at superhelical apices is much greater than the intrinsic curvature of A-tracts in unconstrained linear DNA[118]. Intrinsic curvature also has a minimal effect on the DNA persistence length[119]. Therefore, intrinsic curvature, alone, is insufficient to allow a particular sequence to bend enough to form a superhelical apex. Whether denaturation, curvature, sequence-dependent flexibility, or a combination of these properties is the primary driver for pinning specific sequences at the apical loops of plectonemes remains to be determined. All three factors likely contribute, with denaturation becoming more prevalent, and therefore a bigger influence, with increasing negative supercoiling.

Molecular dynamics simulations help to provide a mechanistic framework for the nucleation and subsequent expansion of denaturation sites[62,65,92,120]. There is an energetic barrier to the initial nucleation event of a base flipping from the helix[62,65]. Therefore, large denaturation bubbles, required for S1 nuclease activity, likely arise from the expansion of shorter Bal-31 sus-ceptible sites, rather than spontaneous formation. Unsurprisingly, the major cleavage hotspots for S1 nuclease corresponded to the ones for Bal-31 but the propensity to expand into large dena-turation bubbles varied among the three sites.

Whereas site 3 was a relatively minor Bal-31 cleavage site, it was a major S1 nuclease cleavage site. Bal-31 cleavage at site 3 is only significant for the less negatively supercoiled topoisomers but becomes suppressed with increased negative supercoiling. The $Lk = 30$ ($\Delta Lk = -2$; $\sigma = -0.064$) topoisomer exhibits a larger fraction of open circular conformations[58], allowing denaturation at sites other than superhelical apices (note, $\sigma = -0.070$ for the $Lk = 30$ topoisomer under the conditions employed in the cryo-ET study). As negative supercoiling increases, and the DNA

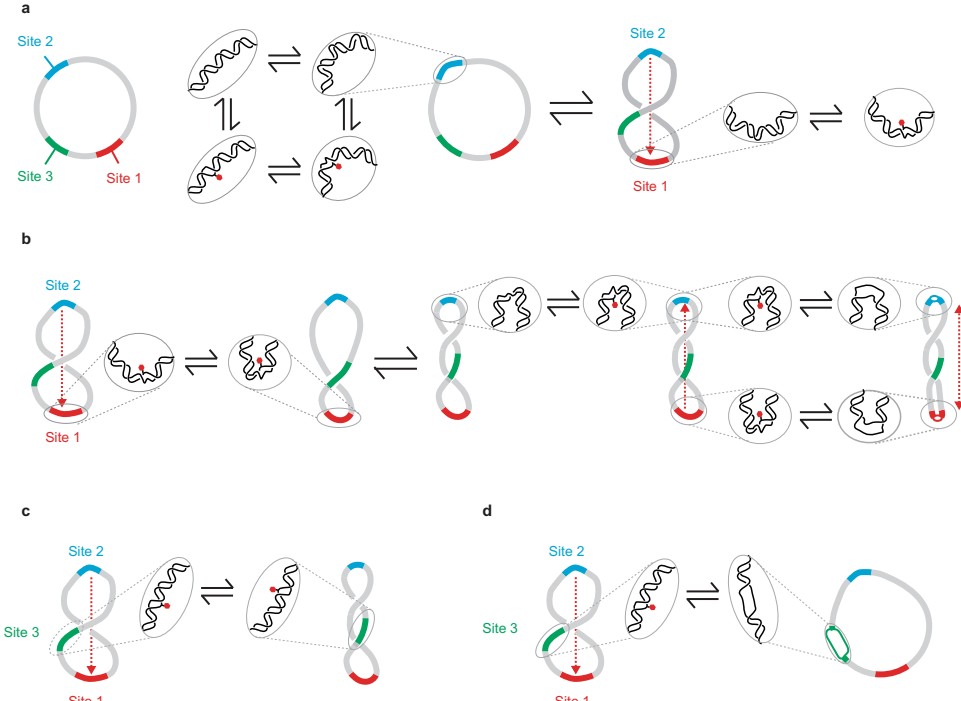

**Fig. 6 Model for cooperative base exposure among distant DNA sites. a** Left, location of Bal-31 and S1 nuclease cleavage sites on the minicircle sequence. Right, sequence-dependent flexibility, intrinsic curvature, and/or base pair disruption induces a bend at site 2. The insets show the details of the local structure of the double helix with flipped-out bases indicated in red. The bend at site 2 is localized to a superhelical apex, placing site 1 at the opposite apex. Increased bending at site 1 leads to disruptions in base pairing. The red dashed line represents the mechanical correlation between the two sites, transferring superhelical strain from site 2 to site 1; the direction of the correlation is indicated by the arrow. **b** Base pair disruption at site 1 introduces a hyperflexible site, allowing the plectoneme loop containing site 1 (red) to become smaller; therefore, facilitating writhe. Increased writhe subsequently reduces the size of the plectoneme loop containing site 2 (blue). The resultant bending strain increases the propensity for base-pair disruption at site 2. With increased negative supercoiling, denaturation sites expand, leading to single-stranded DNA bubbles at the apices. **c**. Site 3 does not appear to be preferentially localized to a superhelical apex. Increased writhe does not affect DNA bending at site 3; therefore, disruption at site 3 probably does not facilitate writhe. **d** Torsional strain relieved by extensive denaturation instead of writhe when base-pair disruption occurs at site 3 instead of sites 1 or 2. Base pair disruption at site 3 would be detrimental to DNA stability.

becomes more highly writhed, nucleation at sites 1 and 2 becomes more favorable whereas nucleation at site 3 is inhibited. Base pair disruption at sites 1 and 2 release torsional strain, thus inhibiting nucleation at site 3. Even though nucleation at site 3 decreases with increasing negative supercoiling, once formed, the subsequent expansion of this site becomes more favorable. In comparison, sites 1 and 2 are much less likely to expand into a large single-stranded bubble. Large segments of single-stranded DNA render DNA more susceptible to damage, are a hallmark of cellular stress, and may cause genetic instability[121]. Therefore, the localization of base-pair disruption to superhelical apices helps to preserve genomic integrity.

A very small stretch of unpaired bases may be sufficient to relieve torsional strain if it occurs at a superhelical apex because the denaturation allows the DNA to writhe more easily. To absorb one negative $Lk$ by denaturation alone, without writhe, ~10 bp of DNA must completely melt. Denaturation at sites 1 and 2 should facilitate writhing, thus contributing to overall stability. Site 3, because it is not localized at an apex, presumably requires a much larger segment of single-stranded DNA to relieve the torsional strain, thus potentially explaining the preferential activity of S1 nuclease at site 3.

The reduced propensity of sites 1 and 2 to form S1 nuclease-susceptible sites, compared to site 3, further supports our idea (postulated above) that the coupling of writhing and denaturation helps to preserve the overall integrity of the DNA molecule. Writhing and denaturation both have considerable nucleation costs, but each process reduces the nucleation cost of the other[112]. Previous observations using single-molecule manipulation of stretching and twisting DNA lend support to our reasoning. Cees Dekker and co-workers[122] made the surprising observation that negatively supercoiled GC-rich DNA melts at lower tensile forces than negatively supercoiled AT-rich DNA. The authors explained this seemingly counterintuitive observation by suggesting that short denaturation bubbles form more readily in AT-rich DNA under torsional strain than in GC-rich DNA. Once formed, these bubbles facilitate DNA writhing, thereby reducing superhelical stress and reducing the likelihood of more extensive denaturation. It should be noted that DNA melting was not directly observed in the single-molecule study but inferred from differences in the DNA extension. Alternative explanations for those differences include Z-DNA[123,124] and L-DNA[125], both of which form more favorably in GC-rich DNA.

Cooperative effects among distant sites and their influence on the 3D conformation of DNA, spatial localization of DNA sequences, and DNA stability are summarized in Fig. 6. This dynamic interplay profoundly influences all interactions involving DNA. A key take-home message of this study is how the coupling of base-pair disruption and writhing contributes to the overall stability of DNA. By dampening the powerful supercoil-induced effects on DNA stability, especially from the hyper-negative dynamic supercoiling generated during transcription, this coupling helps to prevent excessive denaturation. With this information, we can begin to understand how supercoiling allows

DNA to meet the seemingly contradictory requirements to protect the genetic information yet allow access to the bases when required. Our results revealed that localized disruption of base pairing is a common occurrence in supercoiled DNA, and the frequency and extent of these distortions sharply increase when supercoiling exceeds a particular threshold. This threshold is often surpassed in homeostatic conditions in vivo[16,17,21], and certainly exceeded when DNA is being actively transcribed[45,46]. Thus, localized denaturation could be exploited as a regulatory mechanism through changes in the supercoiling level.

Studies of DNA carried out using short duplexes cannot capture the long-range correlations that arise from supercoiling and curvature. Thus, another key take-home message of this study is that to understand how proteins, drugs, and other ligands interact with DNA, one must consider supercoiling, circularity, and long-range mechanical correlations. A recent study showed that ignoring these important variables can have significant consequences. The study revealed that CRISPR-Cas9 nuclease is modulated by DNA supercoiling, which leads to increased cleavage at off-target sites[126]. Similarly, our results with Bal-31 cleavage of nicked minicircles demonstrate how an isolated duplex of DNA does not adequately represent the behavior of topologically constrained DNA loops found in vivo. Constraining DNA into loops increases the distortion at a nick site and thus facilitates recognition and repair of lesions in the genome.

We have shown that nucleases with no sequence specificity can be made to be sequence-specific with supercoiling and curvature. Wilma Olson and coworkers showed a similar effect of long-range mechanical correlations conferring sequence specificity on the non-specific architectural protein, HU, when they simulated DNA being constrained into loops[127]. Similarly, the major hotspot for Bal-31 and S1 nuclease would not exist were it not for the placement of site 2 opposite. Therefore, long-range mechanical correlations communicate the effect of distant sites to transfer superhelical stress.

The minicircle system we developed to study supercoiling-dependent structural changes has additional DNA nanotechnology applications (reviewed in[128]), including for gene therapy[129–132]. DNA nanoparticles (specific shapes formed from annealed single strands of DNA) have been used for targeted drug delivery[133], genome editing[134], diagnostic imaging[135], quantitative biosensing of biomarkers[136], and many additional previously unanticipated applications[137]. Our enhanced understanding of how DNA sequence, supercoiling, and localized disruptions modulate the 3D conformation of DNA suggests that supercoiled minicircle nanoparticles could be designed with specific shapes[56,63], which may prove useful for nanotechnology.

## Methods

**Minicircle DNA**. The 336 bp minicircle DNA used in this study has the following 5′–3′ sequence. The first base listed below is designated as the start of the sequence, and the bases are listed from 1 to 336. Restriction sites, EcoRV, XmnI, BbvCI, NdeI, and MseI, used as sequence markers for the mapping experiments, are underlined and appear in the order listed. Minicircle DNA treated with Nb.BbvCI is nicked at the BbvCI site. The integrase site *att*R is highlighted in bold. Intrinsic curvature in *att*R mapped close to the MseI site (Supplementary Fig. 1), in agreement with previously published data[115]. **TTTATACTAACTTGAGCGAAACG**GGAAGGG TTTTCACCGATATCACCGAAACGCGCGAGGCAGCTGTATGGCATGAAG AGTTCTTCCCGGAAAACGCGGTGGAATATTTCGTTTCCTACTACGAC TACTATCAGCCGGAAGCCTATGTACCGAGTTCCGACACTTTCATTGAGA AAGAT**GCCTCAGCTCTGTTACAGGTCACTAATACCATCTAAGTAGTTGA TTCATAGTGACTGCATATGTTGTGTTTTACAGTATTATGTAGTCTGTT TTTTATGCAAAATCTAATTTAATATATTGATATTTATATCATTTTACG TTTCTCGTTCAGCTTT**. The deletion to generate the 333 bp minicircle DNA removed the underlined bases: TGTAT**GGC**ATGAA to TGTATATGAA. The 339 bp minicircle DNA studied contains the underlined insertion: TGTATGGC ATGAA to TGTATGGC**GAA**ATGAA. Double-length 666 and 672 bp minicircles contain two copies (in tandem orientation) of the 333 and 336 bp minicircles respectively and are generated as byproducts during the recombination process to make 333 and 336 bp minicircles.

**Chemicals and reagents**. Nuclease Bal-31, Nb.BbvCI, BbvCI, NdeI, MseI, EcoRV, XmnI, Exonuclease III, T4 DNA Ligase, Proteinase K, low molecular weight DNA ladder, and 100 bp DNA ladder were purchased from New England Biolabs (Ipswich, MA). S1 nuclease was purchased from Thermo Fisher Scientific (Waltham, MA). Adenosine triphosphate (ATP), dithiothreitol (DTT), DNase I, ethidium bromide, and RNase A were purchased from Sigma-Aldrich (St. Louis, MO). Acrylamide, ampicillin, chloroform, isopropyl beta-D-1-thiogalactopyranoside (IPTG), sodium chloride, and sodium citrate were purchased from Fisher Scientific (Pittsburgh, PA). All other chemicals were purchased from VWR International (West Chester, PA).

**Generation and purification of minicircle topoisomers**. Plasmid pMC336[58] was used as parent plasmid to generate both 336 bp and dimer 672 bp minicircles as described previously[57,58]. Similarly, parent plasmid pMC339-BbvCI[57] was used to generate 339 bp minicircles. Plasmid pMC333, which was used to generate both 333 bp and 666 bp minicircles, was constructed by deleting three base pairs from pMC336 using the QuikChange II site-directed mutagenesis kit (Stratagene, La Jolla, CA).

Negatively and positively supercoiled minicircle DNA topoisomers were prepared as described previously[57,58]. Negatively supercoiled topoisomers were generated by religating nicked minicircles in the presence of ethidium bromide. Positively supercoiled topoisomers were generated by religating nicked minicircles in the presence of HmfB. HmfB was expressed from the plasmid pKS323[138] and purified as described previously[58]. Individual minicircle topoisomers were isolated by preparative polyacrylamide gel electrophoresis (5% polyacrylamide (acrylamide:bis-acrylamide = 29:1) for 333, 336, and 339 bp minicircle topoisomers and 4% polyacrylamide (acrylamide:bis-acrylamide = 29:1) for 666 and 672 bp minicircles. Gels were run at 125 V (~6 V/cm) for 8 h in 40 mM Tris-acetate buffer containing either 10 mM CaCl₂ (for negatively supercoiled topoisomers), 1 mM EDTA (for positively supercoiled topoisomers), or 1 mM EDTA with 2 µg/ml ethidium bromide (for relaxed topoisomers).

Preparative gels were stained with ethidium bromide, DNA bands were excised, and DNA electroeluted from the gel slices at 80 V for ~16 h in D-tube dialyzers (Novagen, Madison, WI) in 40 mM Tris-acetate, 1 mM EDTA (TAE) buffer. Electroeluted DNA was extracted with butanol to both reduce the volume and remove any residual ethidium bromide, extracted with chloroform, precipitated with ethanol, resuspended in 10 mM Tris-HCl pH 8.0, 0.1 mM EDTA (T(0.1)E buffer), desalted using an Amicon 0.5 ml centrifugal filter (EMD Millipore, Billerica, MA), precipitated with ethanol, and resuspended in T(0.1)E buffer. DNA concentrations were determined using a Nanodrop spectrophotometer (Thermo Scientific, Wilmington, DE).

**Generation of nicked and gapped substrates**. Minicircle DNA was nicked at a single site using nicking endonuclease Nb.BbvCI according to the manufacturer's protocol. The DNA was subsequently incubated at 80 °C to inactivate the nicking enzyme, precipitated with ethanol, and resuspended in 10 mM Tris-HCl pH 8.0, 1 mM EDTA (TE) buffer. Gapped DNA was generated by limited digestion of nicked minicircle with Exonuclease III. Nicked minicircle DNA (25 µg) was incubated with 50 units of Exonuclease III at 25 °C. After 2 min of incubation, the reaction was quenched with EDTA (40 mM final concentration). The reaction was incubated at 70 °C for 20 min to inactivate the exonuclease, precipitated with ethanol, and resuspended in T(0.1)E buffer.

Nicked and gapped minicircles were purified by preparative polyacrylamide gel electrophoresis (5% polyacrylamide, acrylamide: bis-acrylamide = 29:1) at 125 V (~6 V/cm) for 8 h in 40 mM Tris-acetate buffer containing 10 mM CaCl₂. DNA was purified and precipitated as above, resuspended in TE buffer, desalted using an Amicon 0.5 ml centrifugal filter, precipitated with ethanol, and resuspended in TE buffer. DNA concentrations were determined using a Nanodrop spectrophotometer.

Bal-31 nicked intermediate was generated by incubating 2.5 µg of the 336 bp, $Lk = 26$, $\Delta Lk = -6$, $\sigma = -0.189$ topoisomer with 2.5 units of Bal-31 in Bal-31 reaction buffer (20 mM Tris-HCl pH 8.0, 600 mM NaCl, 12 mM MgCl₂, 12 mM CaCl₂, and 1 mM EDTA) at 30 °C in a total volume of 1,000 µl. After 1 min, the reaction was quenched by the addition of EDTA (50 mM final concentration), followed by incubation at 65 °C for 10 min to inactivate the enzyme. DNA was subsequently concentrated and desalted using Amicon Ultra 0.5 ml centrifugal filters (Millipore Sigma, Burlington, MA). DNA was recovered in T(0.1)E buffer and purified by preparative gel electrophoresis as described above.

**Calculation of helical repeat**. $Lk = 31$, 32, and 33 topoisomers of the 333, 336 and 339 bp topoisomers were loaded on a 5% polyacrylamide gel (acrylamide:bis-acrylamide = 29:1) and electrophoresed at 30 V (~1.5 V/cm) for 8 h in 40 mM Tris-acetate buffer containing 600 mM NaCl, 12 mM MgCl₂, 12 mM CaCl₂ (to replicate Bal-31 reaction conditions). The gel was subsequently stained with SYBR Gold and the relative electrophoretic mobilities measured using ImageQuant TL (see Source Data). These mobilities were plotted against $Lk$. For the 333 bp minicircle, a line was plotted through the datapoints for the $Lk = 32$ and 33 topoisomers. Relative electrophoretic mobilities are assumed to follow a V-like dependence on $\Delta Lk$[80,139], therefore, a line with equal and opposite slope was fitted

through the $Lk = 31$ topoisomer. Where the two lines intersect is estimated to be the $Lk_0$. This was repeated for the 336 and 339 np minicircles to estimate $Lk_0$, except that the first line was fitted through the $Lk = 31$ and 32 topoisomers. The mean helical repeat (averaged from the three different minicircles) was estimated to be 10.48 (±0.02) bp/turn. Values for $Lk_0$ for each substrate were calculated using this value. The helical repeat under S1 nuclease conditions was not determined; however, it should be similar to the value under Bal-31 reactions, and we, therefore, used the same value of ~10.48 bp/turn. The linking number difference ($\Delta Lk$) is calculated from $\Delta Lk = Lk - Lk_0$, this is scaled to the size of the DNA to give the superhelical density ($\sigma$) = $\Delta Lk/Lk_0$.

**Mapping of Bal-31 cleavage sites.** 2.5 μg of each topoisomer was incubated with 2.5 units of Bal-31 in Bal-31 reaction buffer at 30 °C in a total volume of 1,000 μl. After 1 min, the reaction was quenched by the addition of EDTA (50 mM final concentration), followed by incubation at 65 °C for 10 min to inactivate the enzyme. For the more slowly cleaved $Lk = 31$ ($\Delta Lk = -1$; $\sigma = -0.033$) topoisomer, Bal-31 (2.5 units) was first incubated with proteinase K (0.68 units) at 37 °C in Bal-31 reaction buffer (total volume of 200 μl) to selectively attenuate the exonuclease activity of Bal-31[140] prior to incubating with the DNA. 2.5 μg of the $Lk = 31$ ($\Delta Lk = -1$; $\sigma = -0.033$) topoisomer was then incubated with 2.5 units of the proteinase K-treated DNA for 10 min at 30 °C. At the end of the incubation, the reaction was quenched with EDTA and the samples treated as for the other topoisomers. Control experiments with the $Lk = 29$; $\Delta Lk = -3$ topoisomer confirmed that the proteinase K treatment did not alter the site of Bal-31 cleavage.

In the previous study[58], the full-length linear intermediate was gel-purified to facilitate the analysis, but this gel-purification may introduce bias by selecting for the DNA in the major band. To eliminate the potential bias, we omitted the gel-purification step in the current study. Bal-31 cleaved DNA was concentrated and desalted using Amicon Ultra 0.5 ml centrifugal filters. DNA was recovered in T(0.1)E buffer. 100 ng of Bal-31 cleaved DNA was incubated with the appropriate restriction enzyme at 37 °C in the buffer recommended by the manufacturer. Reactions were quenched by the addition of EDTA (20 mM final concentration) and incubated at 65 °C for 20 min. Products were analyzed by electrophoresis on 3% agarose gels (SeaKem LE agarose, Lonza, Rockland, ME) at 100 V for 3 h. Following electrophoresis, gels were stained with SYBR Gold, visualized using a FOTO/ANALYST Investigator imaging system (Fotodyne, Hartland, WI), and analyzed using ImageQuant TL (GE Healthcare Life Sciences, Marlborough, MA) software. Fragment lengths in bp were determined using the molecular size calibration feature of ImageQuant TL, with the low molecular weight DNA ladder as the standard. Multiple different fragment lengths were used to determine each cleavage site taken from different starting points (restriction sites) and in different directions. These results were averaged to reduce the noise and determine the error. 100 ng samples of each Bal-31-cleaved topoisomer subsequently cleaved with BbvCI or NdeI were also loaded on 5% polyacrylamide gels (TAE buffer). Gels were run at 125 V for 4 h, subsequently stained with SYBR Gold, then visualized using a FOTO/ANALYST Investigator imaging system. Band volumes were quantified using ImageQuant TL. Cleavage frequencies were determined from the molar ratio of the product fragments, calculated by comparing band volumes. SYBR Gold staining is proportional to the total mass of DNA; therefore, band volumes were corrected for the difference in fragment sizes to obtain the molar ratios.

**Mapping of S1 nuclease cleavage sites.** S1 nuclease cleavage sites were mapped in a similar manner as for Bal-31 cleavage sites; however, S1 nuclease generated mostly nicked products (without gaps) when incubated with negatively supercoiled minicircles. To facilitate the comparison with Bal-31, the nicked products generated from S1 nuclease activity on negatively supercoiled minicircles, were incubated with additional S1 nuclease at a higher temperature (see below), to convert the nicked products to linear DNA. S1 nuclease cleaves DNA endonucleolytically on the strand opposite a nick[69,141]. The activity of S1 nuclease on nicked minicircle DNA was relatively inefficient, requiring increased enzyme concentration, temperature, and incubation time to allow S1 nuclease to cleave the strand opposite the previously generated nick. Specifically, 2.5 μg of each topoisomer was incubated with 250 units of S1 nuclease in S1 nuclease reaction buffer (40 mM sodium acetate pH 4.5, 300 mM NaCl, 2 mM ZnSO$_4$) at 30 °C for 1 h in a total volume of 1,000 μl. Subsequently, an additional 375 units of S1 nuclease was added, the incubation temperature was increased to 37 °C, and the reaction incubated for an additional 2 h to allow for S1 nuclease to convert the nicked intermediate to linear DNA. At the end of the incubation, the reaction was quenched by the addition of 100 μl 500 mM EDTA, followed by incubation at 70 °C for 10 min to inactivate the enzyme. S1 nuclease-cleaved DNA was then concentrated, desalted, and analyzed following the same protocol used for Bal-31 cleaved DNA described above.

**Kinetic analysis of DNA cleavage by nuclease Bal-31.** Minicircle DNA (150 ng in 60 μl final volume) was incubated with 0.15 units of nuclease Bal-31 at 30 °C in Bal-31 reaction buffer. To eliminate any batch-to-batch variations in Bal-31 activity, each comparison of relative cleavage rates was performed using enzymes from the same batch. At appropriate time points (determined empirically by preliminary kinetic analyses), 25 ng samples were removed and quenched by the addition of an equal volume of stop buffer (50 mM Tris-HCl pH 8.0, 100 mM disodium EDTA, 200 μg/ml

proteinase K, and 10% glycerol). Samples were subsequently incubated at 45 °C for 30 min to allow for the proteinase K in the stop buffer to degrade Bal-31, followed by an additional incubation at 65 °C for 20 min to ensure the enzyme was completely inactivated. The reaction products were analyzed by electrophoresis on 5% polyacrylamide gels (acrylamide: bis-acrylamide, 29:1) in TAE buffer. Gels were run for 4 h at 125 V (~6 V/cm) with continuous buffer recirculation. Following electrophoresis, gels were stained with SYBR Gold and visualized using a FOTO/ANALYST Investigator imaging system (Fotodyne, Hartland, WI) with quantification initially employing TotalLab software (TotalLab, Newcastle, U.K.), and in later work using ImageQuant TL (GE Healthcare Life Sciences, Marlborough, MA), an updated version of the TotalLab software.

Bal-31 steady-state cleavage rates were determined by quantifying the disappearance of the supercoiled substrate over time. Data were plotted as the uncut fraction against time using KaleidaGraph (Synergy Software, Reading, PA) and fitted to straight lines to determine the cleavage rates. All cleavage rate evaluations were performed at least three times from which mean rate constants and standard deviations were calculated.

**Kinetic analysis of DNA cleavage by S1 nuclease.** Minicircle DNA (150 ng in 60 μl final volume) was incubated with 15 units of S1 nuclease at 30 °C in S1 nuclease reaction buffer. Although S1 nuclease is active at neutral pH in the presence of Mg$^{2+}$ ions[142], the reaction required much higher enzyme concentrations and longer incubation times compared with reactions at acidic pH. Consequently, the standard reaction buffer (pH 4.5) was used. At appropriate time points (determined empirically by preliminary kinetic analyses), 25 ng samples were removed and quenched by the addition of an equal volume of stop buffer (50 mM Tris-HCl pH 8.0, 100 mM disodium EDTA, 200 μg/ml proteinase K, and 10% glycerol). Samples were subsequently incubated at 45 °C for 30 min to degrade S1 nuclease, followed by an additional incubation at 70 °C for 10 min to ensure the enzyme was completely inactivated. The reaction products were analyzed by electrophoresis and reaction rates were determined using the same protocol as used for measuring the kinetics of Bal-31 cleavage.

**Reporting summary.** Further information on research design is available in the Nature Research Reporting Summary linked to this article.

## Data availability
The data that support this study are available from the corresponding author upon reasonable request. Gels used to map enzyme cleavage sites are included in Supplementary Figs. 4, 5 and 6. Source data are provided with this paper.

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

## Acknowledgements

They dedicate this work to the memory of Dr. Jorge Bernardo Schvartzman, a friend and deep thinker who inspired the field. We thank Dr. Daniel J. Catanese, Jr. for critically reading the paper, Dr. Michelle C. Swick for constructing the pMC333 plasmid, Dr. Nancy Crisona (then of the University of California at Berkeley), and Dr. Kathleen Sandman (The Ohio State University) for the HmfB expression plasmid, pKS323, and invaluable advice on the purification of HmfB and its utility for making positively supercoiled DNA. This work was supported in part by National Institutes of Health (NIH) grants R56 AI054830 and R01 GM115501 to L.Z.

## Author contributions

J.M.F. designed and performed most of the experiments. L.Z. helped to design the experiments and assisted with data analysis. A.K.J., E.S., and H.L.C. assisted with nuclease cleavage site mapping experiments and helped to analyze data. Figures were prepared by J.M.F. All authors contributed to writing the paper.

## Competing interests

J.M.F. and L.Z. are co-inventors on several patents covering the minicircle technology and are shareholders in Twister Biotech, Inc. The authors declare no other competing interests.
