## [Peer Review File · Nature Communications]

Supercoiling and looping promote DNA base accessibility and coordination among distant sitesREVIEWER COMMENTS

Reviewer #1 (Remarks to the Author):

This study by Fogg et al. builds on their previous work which showed by Cryo-ET that negative supercoiling in constrained DNA results in a wide variety of DNA conformations, which contain highly bent regions, likely to be the site of local disruptions to DNA base-pairing. In this study Fogg et al map the sites of exposed bases in supercoiled DNA minicircles and quantify the levels of disruption occurring in these regions, using Bal-31 and S1 nuclease. Their findings show that local distortions to the DNA structure are the root cause of the variety in structure previously observed.

A secondary but extremely important finding of this study is that Bal-31 cleaves DNA at different rates, depending on the level of disruption at the processing site. This finding will be of significant use to the wider field. Fogg and coworkers use this increased understanding to demonstrate that in addition to the level of disruption, the rotational alignment of DNA bases, can have a significant effect on the enzymatic processing of disruptions by changing the number of base pairs in their substrates to increase rotational misalignment.

Finally, Fogg and coworkers show cooperative effects between sites at opposite ends of the DNA minicircle structures, and that this cooperativity can result in changes to the sites of disruption for different DNA minicircle sequences. They demonstrate that in DNA minicircle dimers, Bal-31 preferentially cleaves different sites to those observed in the monomeric substrates (site 2 > site 1). By mapping the location of exposed bases and the enzymatic activity of those sites, Fogg and coworkers are able to build a model for how local DNA disruptions act to change the overall conformation of the molecule.

This study therefore provides new and essential insight into the fundamental structures which cause conformational diversity in supercoiled DNA. It also shows that non-sequence specific enzymatic activity can become highly sequence specific in supercoiled DNA as a result of these structural changes. This has downstream ramifications for all DNA interactions, and for the design of DNA nanostructures.

I have the following minor comments:

From figure 3 (D,E), there seems to be a strong transition in enzymatic activity between LK 29 \diamond LK 28, with a large increase in activity of Bal-31 at site 2 and a decrease in S1 nuclease at site 1. From figure 4B, this also seems to correlate to a measurement with smaller error bars – is there therefore a significant difference in the number of events at this point that could be causing this shift? If so, might it be useful to show N in addition to the fraction at each site? This figure in general seems to be more information dense than the accompanying discussion. It would be interesting to see models as shown in figure 6 accompanying these changes in binding site preference, to aid in the interpretation of what structural changes are driving/driven by these. For plots with fits, it would be useful if the calculated rates and errors were shown in the captions.

Reviewer #2 (Remarks to the Author):

The manuscript by Fogg et al, "Supercoiling and looping promote DNA base accessibility and coordination among distant sites" describes studies of the nature and locations of disruptions of B-DNA double helix that are induced by different levels of supercoiling in DNA mini-circles. The authors note that such mini-circles model loops in DNA that are prevalent in intracellular genomes. The authors build on an earlier study using electron cryotomography of the morphology of DNA circles ~330 bp in length (varied slightly to generate integral or fractional numbers of relaxed helical turns) and subjected to a range of torsional stress, from highly underwound to highly overwound. In this previous study DNAs were sorted into several interconvertible classes and local disruptions of the double helix were inferred from the trajectories of the DNA backbone. Here the integrity of the DNA helix is interrogated using Bal31 and S1 nucleases. To accomplish this, the authors perform careful kinetic studies that show that the rapid endonuclease activity of Bal31 is better than S1 at detecting single-nucleotide disturbances and that with short incubations, this activity can be distinguished from the former enzyme's double-strand exonuclease action. The

authors demonstrate that the helical phasing of the mini-circles determines their sensitivity to Bal31. They use restriction enzyme digestion and nicking-restriction enzymes to map the sites of Bal31 cleavage. They eliminate effects of intrinsic curvature on the size estimation of the fragments by using agarose gels after making the interesting observation that such mobility differences, readily apparent on polyacrylamide gels, are minimal on agarose. They show that there are 3 preferred sites of cleavage that are used differentially according to the level of negative supercoiling. Although only one site is predicted to be a sequence that melts in response to supercoiling, they assert that buckling at this site is necessary to enable melting and sharp bending at the predicted site on the opposite side of the ring. The authors also show that only with high levels of stress does positively supercoiled DNA become sensitive to Bal31 suggesting that formation of a focus of a distinct non-B DNA structure, likely P-DNA. The authors next perform similar experiments using dimers of the mini-circles. Due to the extra length, these circles can absorb relatively more stress as writhe and so have less propensity to unwind bases. The authors discuss the considerable implications of this study on DNA damage recognition by repair machinery and on genetic transactions and chromatin in general.

This is a well-written, well-executed and interesting scholarly study with important implications for all DNA processes. The principles demonstrated in, and derivable from the work in this manuscript will have implications for the design of lentivectors, CRISPR systems, and for the general understanding of gene expression, DNA repair and recombination.

Nevertheless, I have several suggestions that if incorporated I believe would improve the manuscript.

1. The authors use electrophoretic methods to map the sites of Bal31 cleavage. I think that it should be relatively easy to old-fashioned sequencing or primer extension or end-labeling experiments, etc. to identify to the exact sites of cleavage of Bal31 and S1. This is important to validate the WebSIDD, molecular dynamics, or other algorithmic predictions. Such validation is important if the findings in this manuscript were to be incorporated into a genomic toolkit.
2. The literature surrounding the detection of single mismatches or nicks with these enzymes is complicated with conflicting reports and a lot of anecdotal evidence ever since Schenk et al, PNAS 72:989-993. The authors might want to consider designing a few nicked, gapped, mismatched or double-stranded oligonucleotides for use as substrates to precisely demonstrate the exact cleavage sites and the extent of cleavage for Bal31 or S1-nucleases in unstressed sequences around the susceptible sites. There may be a lot of variation depending on assay conditions, local sequence environment, and topological state.
3. Besides torsional stress, in such sharply bent rings, helical phasing can shift a cleavage site from the interior concave surface of the double helix to an exterior convex surface. I imagine that this may be particularly important for the secondary cleavage sites. Could the authors comment on this?
4. It's interesting to note that S1-mung bean sensitivity was provoked by lac-repressor binding with plac5 (lac operator embedded in lambda DNA almost five decades ago (Chan HW & Wells RD Nature volume 252, pages205–209, 1974). As it was later determined that the repressor loops between lac O1 and O3, this may be an early example of how proteins control base accessibility through looping and how binding affinities can be tuned by topology.
5. The authors indicate that to enable a shape change, that a second helical disruption must occur 180-degrees opposite the first disruption. But they never note explicitly for readers who may not have considered this before, that a rigid ring of any material can never be disrupted at just a single point (indeed in radiology there is principle that whenever there is one break in ring of bone, there is always a second that needs to be located). However, the second helical disruption need not be 180-degrees opposed as asserted. If the mini-circle is allowed to writhe the second disruption may occur anywhere throughout the backbone as puckering pucker relieves the stress.

Reviewer #3 (Remarks to the Author):

Fogg et al. report a series of very interesting studies of the actions of two nucleases (Bal-31 and S1) on a series of DNA minicircles of precise topologies, ranging from highly underwound topoisomers with a linking number difference ΔLk of -6 to overwound topoisomers with a ΔLk of $+3$ and found in previous cryogenic electron tomography studies from this group to adopt a broad and unexpected range of spatial configurations. The enzymes are known from decades of biochemical research to recognize different conformational features of DNA, namely single-nucleotide distortions vs. longer stretches of single-stranded DNA. The interactions of the enzymes with the various topoisomers provide new information on how local distortions might be distributed in the DNA and how they might contribute to long-range allostery.

Unfortunately, the paper is very difficult to read in its current form. The interweaving of structural rationales with experimental results is highly confusing and the interpretation of data is frequently unclear.

The reader does not know why the authors make assorted claims about the data, e.g.:

Why does a base need to be exposed to cut a phosphodiester bond? p. 8 - the phosphodiester linkages are already highly exposed on double helical DNA.

How does one know that the base pair at a nick site on minicircle are unaligned? p. 11 – bases stack one above the other even in small single-stranded hairpin loops and nicked minicircles of 336 bp are not highly constrained (only $360^\circ/336$ bp or $\sim 1^\circ$ if uniformly bent).

How can one claim that bases are exposed on positively supercoiled DNA? p. 17 – the overtwisted DNA on nucleosomes does not show such behavior.

How is it known from Figure S1 that a small fraction of intermediates lack a gap and that there is variability in gap lengths?

In other words, given that these DNA-protein systems are not well characterized at a detailed structural level, it hard to draw firm conclusions about DNA distortions within supercoiled minicircles.

The reader needs guidance in understanding how one can draw conclusions about nuclease cleavage sites from the restriction enzyme treatments, e.g., how one deduces sites 1-3 from the gel patterns and what nucleotides constitute each site. One can't always tell which figure is being discussed when conclusions are drawn – a figure in the text versus a figure in supplementary information. For example, the statement about gap sizes does not follow from Figure S1 (p. 8, line 6 from bottom of page). The logic behind other statements is also not obvious, e.g., p. 14 – lines 4-5 from bottom of page, p. 16 – ¶ 2 lines 4-5, p. 25 – ¶2, lines 5-7.

It would be very helpful if all the sequence-related information about the minicircle (from the methods section and figures) were consolidated so that the reader could see the precise locations of enzyme cutting sites, nuclease recognition sites, attR curvature fragment, Nb.BbvCI site, etc. at the base-pair level. Such information might lead the reader to a better understanding of the logic used in the interpretation of data and appreciation of the information that has been gleaned from this research. An introductory image might also help the reader appreciate the problem under investigation and get a better understanding of the jargon used in describing the configurations of supercoiled DNA.

The casual presentation is sometimes misleading. For example, the definition twist is imprecise (p. 4) – two pathways, not three as suggested in the text, define twist. Linking number is not a term (p. 4). Persistence-length DNA is not essentially rigid (p. 5) – rigid DNA is about an order of magnitude smaller than the persistence length (10's of bps such as the gold-nanoparticle labeled fragments studied by Per Harbury).

Other points:

The cooperative effects observed experimentally by Kim et al. (Probing allostery through DNA. Science 2013, 339, 816-819) are relevant to the discussion on p. 6. The system, while linear, constrained by fixed ends.

Figure 2B - the rotational alignment is not clear

Figure 6A – typo “stain”

Figure 6B – not clear which loop is described as the smaller plectonemic loop

Figure S3 – Michaelis-Menten analysis unclear (symbols not defined)

Reviewer 1:

From figure 3 (D,E), there seems to be a strong transition in enzymatic activity between LK 29 \diamond LK 28, with a large increase in activity of Bal-31 at site 2 and a decrease in S1 nuclease at site 1. From figure 4B, this also seems to correlate to a measurement with smaller error bars – is there therefore a significant difference in the number of events at this point that could be causing this shift? If so, might it be useful to show N in addition to the fraction at each site? This figure in general seems to be more information dense than the accompanying discussion. It would be interesting to see models as shown in figure 6 accompanying these changes in binding site preference, to aid in the interpretation of what structural changes are driving/driven by these.

RESPONSE: What a keen observation and we thank the reviewer for this deep analysis of these admittedly information-dense figures. Although the different topoisomers were cleaved at different rates by S1 nuclease, the mapping experiment was designed such that all of the supercoiled substrate would be nicked during the initial incubation with S1 nuclease. This nicking reaction completion is true even for the more slowly cleaved topoisomers. Thus, there is no difference in the total number of events. For Bal-31, all the negatively supercoiled topoisomers except the $Lk = 31$ topoisomer were cleaved very rapidly. Therefore, there should be no significant difference in the total number of events.

Once the DNA is initially nicked by either Bal-31 or S1 nuclease, the supercoiling is lost. Thus, Bal-31 and S1 nuclease should act only once on each individual minicircle. For example, cleavage at site 1 will prevent subsequent cleavage at sites 2 or 3. The enzyme concentration was much lower than the DNA concentration, therefore we can exclude the possibility of multiple enzymes acting on a single minicircle molecule. We have added this explanation to the revised manuscript. We have also added a schematic as panel (f) of Figure 3 showing how sites 1 and 2 are hypothesized to localize to the apices to aid in the interpretation, as suggested.

For plots with fits, it would be useful if the calculated rates and errors were shown in the captions.

RESPONSE: As per the instructions to authors for this journal, results and data should not be included in figure legends. For Figure 4, the graphs summarize the results from 31 individual cleavage rate measurements. There is not space to list all 31 rates and errors in the figure legend or on the figure. Instead, the calculated rates and errors are listed in Supplementary Table 1. We realized, upon reading this helpful comment, that we had failed to point this fact out to the reader in the previous version of the manuscript. We agree that it is useful to see the actual calculated rates and errors used to generate the figures. To make these data easier to find, we have now added text in both the figure legends and the main text of the revised version of the manuscript that the rates are listed in Supplementary Table 1.

Reviewer 2:

The authors use electrophoretic methods to map the sites of Bal31 cleavage. I think that it should be relatively easy to old-fashioned sequencing or primer extension or end-labeling experiments, etc. to identify to the exact sites of cleavage of Bal31 and S1. This is important to validate the WebSIDD, molecular dynamics, or other algorithmic predictions. Such validation is important if the findings in this manuscript were to be incorporated into a genomic toolkit.

RESPONSE: We felt the same way! And we tried several of these methods, including conventional Sanger sequencing but there was too much noise (to signal). The samples are just too heterogenous. Each topoisomer is cleaved at multiple locations (sites 1, 2, and 3). Control experiments in which we sequenced minicircles cleaved at a single known site (e.g., the EcoRV site) worked well (by Sanger sequencing). The mapped cleavage site sequences were highly varied and widely spread (by approximately 10 bases). One challenge is that Bal-31 makes a gap of varied length (up to ~ 10 bp), following the initial strand cleavage, meaning that there is no nucleotide level sequence in the vicinity of the initial cleavage site to map. We were unable to find any way to stop Bal-31 at the initial point of cleavage. Therefore, it is impossible to determine the exact site of cleavage (to the base pair level). Another challenge is that Bal-31 does not appear to cleave at an exact sequence, but at multiple closely spaced sites within a ~ 20 bp region (note the appearance of multiple bands and smeary bands in Supplementary Figure 6). Mapping S1 nuclease cleavage gave us the same challenge.

The S1 cleavage event falls anywhere within a >4 bp bubble, making it impossible to map the cleavage site to base pair resolution.

Our electrophoretic method, however, provides an accurate measure of DNA lengths to within ~10 bp resolution. The advantage of the electrophoretic method lies in its simplicity and ease of interpretation. The distinct pattern of fragments observed (for example, see Figure 3b), clearly indicates preferential cleavage at a particular site. It is also highly quantitative, allowing us to very accurately determine the relative cleavage at each site. Because of the importance of this issue, we carried out additional analysis to define the range of cleavage sites and quantify the error in the measurement. We also now report mapping results for each individual topoisomer. These new analyses are listed in the new Supplementary Table 2 (for Bal-31) and Supplementary Table 3 (for S1 nuclease). We also added several sentences to the revised version of the paper explaining the new enhanced analysis and a sentence explaining that we were unable to determine the cleavage to base pair resolution and why. We thank the reviewer for prompting us to add this valuable information to the revised manuscript.

The literature surrounding the detection of single mismatches or nicks with these enzymes is complicated with conflicting reports and a lot of anecdotal evidence ever since Schenk et al, PNAS 72:989-993. The authors might want to consider designing a few nicked, gapped, mismatched or double-stranded oligonucleotides for use as substrates to precisely demonstrate the exact cleavage sites and the extent of cleavage for Bal31 or S1-nucleases in unstressed sequences around the susceptible sites. There may be a lot of variation depending on assay conditions, local sequence environment, and topological state.

RESPONSE: Bal-31 has potent exonuclease activity in addition to structure-specific endonuclease activity. Oligonucleotide substrates have free DNA ends and therefore would be rapidly degraded by the exonuclease activity of Bal-31, making them inappropriate substrates. As a control experiment, we mapped the S1 nuclease site on minicircles nicked with Nb.BbvCI (therefore we know exactly where the nick is located in the minicircles). The nicking site was mapped following the same protocol used to map the S1 nicking site in the supercoiled topoisomers, and we verified that, as expected, S1 nuclease cleaved at the site where the DNA was initially nicked (Supplementary Figure 5a). Bal-31 activity on nicked minicircle is too slow to map the cleavage site because of the exonuclease activity of the enzyme; any linearized intermediate is degraded before it can accumulate. We were able to test gapped minicircles and found these to be cleaved very rapidly by Bal-31 to generate linear DNA (See Supplementary Figure 3); we did not map the cleavage site.

Just to be clear: we have no mismatches in any of our sequences. In our experiments, minicircle length and supercoiling are the only variables changed; all other conditions are identical. We also used the same batch of enzyme for each set of experiments. We agree with the reviewer that we would expect different assay conditions, local sequence environment, etc. to also affect nuclease cleavage of minicircles, which is why we kept everything consistent. The sequence in the immediate vicinity of the nick site is unchanged between the different length minicircles. From the 336 bp minicircle, the added three (339 bp) or subtracted three (333 bp) base pairs are located more than 100 bp away from the BbvCI site (where Nb.BbvCI acts) and, therefore, should not affect nuclease cleavage of the nicked site. Instead, our data are consistent with differences in phasing caused by changing DNA length in non-helical repeat (10.5 bp) lengths. Because the reviewer raised this important issue, we have added a few sentences to help clarify these points to readers. We also improved the diagram on Figure 2 (to the right of panel a) to more clearly depict the helical phasing differences.

Besides torsional stress, in such sharply bent rings, helical phasing can shift a cleavage site from the interior concave surface of the double helix to an exterior convex surface. I imagine that this may be particularly important for the secondary cleavage sites. Could the authors comment on this?

RESPONSE: This is an interesting idea, and we agree that helical phasing likely has an effect. In molecular dynamics simulations of ≤100 bp minicircles, the register of DNA kinks is important (Lankas et al. (2006), *Structure* vol. 14, 1527–1534; Mitchell et al. (2011) *Nucleic Acids Research*, vol. 39, 3928–3938). These kinks involve bending towards the minor groove; therefore, they will preferentially form when the minor groove is on the inside of the circles. Thus, the location of the first kink will affect the location of where the second kink will form. Some of these kinks include base pair disruptions. Therefore, helical phasing may affect how and where secondary Bal-31 cleavage sites develop. Our 336 bp minicircles are far more flexible but it is nonetheless possible that a shift from interior to exterior could “reveal” or “hide” nuclease cleavage sites with supercoiling.

In the molecular dynamics simulation studies cited above, disrupted base pairs are typically on the outside of the circle. We have added a couple of sentences to discuss this idea and thank the reviewer for the suggestion.

It's interesting to note that S1-mung bean sensitivity was provoked by lac-repressor binding with plac5 (lac operator embedded in lambda DNA almost five decades ago (Chan HW & Wells RD Nature volume 252, pages205–209, 1974). As it was later determined that the repressor loops between lac O1 and O3, this may be an early example of how proteins control base accessibility through looping and how binding affinities can be tuned by topology.

RESPONSE: We thank the reviewer for their reminder of this important paper and decided to carefully reread it. Although the authors reported that S1 nuclease and mung bean nuclease modulated Lac repressor binding, we did not see anything in the paper that suggested that Lac repressor binding provoked nuclease sensitivity. In contrast, it dampened nuclease activity at some sites in the *lac* operon but did not affect the overall extent to which S1 nuclease cleaved the lambda DNA (see Table 1 of that paper). After careful reading and re-reading, we decided not to add this reference, although we agree it was an important finding that provided early hints of how topology can affect protein binding. It is just a little too diverged from the focus of our paper.

The authors indicate that to enable a shape change, that a second helical disruption must occur 180-degrees opposite the first disruption. But they never note explicitly for readers who may not have considered this before, that a rigid ring of any material can never be disrupted at just a single point (indeed in radiology there is principle that whenever there is one break in ring of bone, there is always a second that needs to be located). However, the second helical disruption need not be 180-degrees opposed as asserted. If the mini-circle is allowed to writhe the second disruption may occur anywhere throughout the backbone as puckering pucker relieves the stress.

RESPONSE: What an interesting observation! We were not aware of the conjecture that a rigid ring of material cannot be broken at a single point. Unfortunately, even after much searching, we were unable to learn any more about this concept. We appreciate the suggestion but are unable to incorporate the idea into the manuscript.

Reviewer 3:

Unfortunately, the paper is very difficult to read in its current form. The interweaving of structural rationales with experimental results is highly confusing and the interpretation of data is frequently unclear.

RESPONSE: In response to this comment, we have carefully gone over the revised manuscript to improve the readability and to present experimental results first and possible structural rationale second. We hope the revised version is improved. We recognize that this is a tricky and challenging story and we worked diligently to make it clear.

Why does a base need to be exposed to cut a phosphodiester bond? p. 8 - the phosphodiester linkages are already highly exposed on double helical DNA.

RESPONSE: Evidence in previously published data from others cited in our paper strongly suggests that both Bal31 and S1 nuclease recognize exposed bases—single exposed bases for Bal31 and a bubble of >4 exposed bases for S1 nuclease. Bal-31 and S1 nuclease recognize the exposed bases but act by cleaving a phosphodiester bond. Perhaps it is not that the base is exposed *per se* but the resulting stress/pucker on the phosphodiester bond resulting from the flipped out exposed base that is recognized by nucleases. Relaxed circular DNA is not cleaved by either nuclease because it contains no exposed bases.

How does one know that the base pair at a nick site on minicircle are unaligned? p. 11 – bases stack one above the other even in small single-stranded hairpin loops and nicked minicircles of 336 bp are not highly constrained (only 360°/336 bp or ~1° if uniformly bent).

RESPONSE: Indeed, because of the even number (32) of helical turns in the nicked 336 bp minicircle, the base pairs flanking the nick site are in a favorable rotational alignment and, consequently, this nicked minicircle is a very poor substrate for Bal31 nuclease. It is only when base pairs are added or subtracted do the base pairs flanking the nick site become unaligned in DNA nicked by Nb.BbvCI nicking endonuclease. Bases likely would be less well stacked at the nick site in the out of phase minicircles because of the lack of rotational alignment. The alignment depends upon the helical phase. We have added additional explanation in the revised manuscript. We also added diagrams to the right of Figure 2a to better illustrate the rotational alignment at the nick.

How can one claim that bases are exposed on positively supercoiled DNA? p. 17 – the overtwisted DNA on nucleosomes does not show such behavior.

RESPONSE: What we claim in the manuscript is only that positively supercoiled DNA is a substrate for Bal-31. That is the experimental result. Our structural rationale for this Bal31 activity on positively supercoiled DNA is that bases are exposed (given previously published results that show that exposed bases is what Bal31 recognizes). We have tightened how we wrote this section in the revised version of the manuscript to clearly separate results from interpretation.

How is it known from Figure S1 that a small fraction of intermediates lacks a gap and that there is variability in gap lengths?

RESPONSE: Upon rereading the manuscript with this comment in mind, we realized that we did not adequately describe how we concluded that there is variability in gap lengths. We have extensively revised the relevant section to better explain our logic. This revision involved reordering the Supplementary Figures. What was Supplementary Figure 1 is now Supplementary Figure 3. The arrangement of the panels in what is now Supplementary Figure 3 has also been revised to make it easier for the reader to follow. Furthermore, in the revised manuscript, instead of simply referring to Supplementary Figure 3, we now explain our reasoning. We thank the reviewer for this question.

In what is now Supplementary Figure 3, the evidence for a small fraction of intermediates lacking a gap comes from the ligase assay. We used two controls. The first, Nb.BbvCI-nicked minicircle, is nicked at a single site and does not contain a gap. T4 DNA ligase successfully repaired the nick and the ligation products are obvious on the gel. The second control, gapped minicircle could not be ligated and no ligation products were observed. It is also known from previous work that T4 DNA is extremely inefficient ligating across a gap (Nilsson and Magnusson (1982) *Nucleic Acids Research*, vol. 10, 1425-1437). We have added this reference to the revised manuscript. Following incubation of the Bal-31 intermediate with T4 DNA ligase, some ligation products were observed, but most of the intermediate was not ligated. From comparison with the controls loaded either side of the Bal-31 intermediate, we conclude that the small fraction of the Bal-31 nicked sample that can be ligated must lack a gap. Gapped minicircle cannot be efficiently ligated.

Further evidence comes from the migration of the nicked and gapped intermediates on the gel. Nb.BbvCI-nicked minicircle migrated more slowly than the gapped minicircle. For minicircles nicked by Bal-31 there is a resulting DNA band that migrates with the Nb.BbvCI-nicked minicircle (thus it is likely singly nicked) but there are additional DNA product bands that run faster, which likely contain gaps. These are not single bands but are two bands with a DNA smear. We have tightened this section and hope it is clearer in the revised manuscript. The band in the Bal-31 intermediate with similar migration to the Nb.BbvCI-nicked minicircle is almost completely absent in the sample taken after the ligase incubation, strongly suggesting that the DNA in the band lacked a gap and was thus able to be ligated. The evidence for a variety of gap lengths comes from both the migration of the Bal-31 intermediate on the gel and the fact that the degradation of the Bal-31 intermediate is multiphasic. It is also known from previous literature cited in the manuscript that Bal31 makes a variety (1 to 10 bp) of gap lengths. Our data support this earlier work.

In other words, given that these DNA-protein systems are not well characterized at a detailed structural level, it hard to draw firm conclusions about DNA distortions within supercoiled minicircles.

RESPONSE: We agree and have now demarcated results from interpretations throughout the revised manuscript.

The reader needs guidance in understanding how one can draw conclusions about nuclease cleavage sites from the restriction enzyme treatments, e.g., how one deduces sites 1-3 from the gel patterns and what nucleotides constitute each site.

RESPONSE: The cleavage site mapping is based upon comparing the lengths of DNA generated when first cleaved with Bal31 or S1 nuclease and then cleaved with one of several different restriction endonucleases. Because the precise location of the cleavage by these latter enzymes are known, the length of the resulting double-cleaved DNAs tells the location of where Bal31/S1 nuclease must have cleaved. The direction from the Bal31/S1 nuclease cleavage is determined from the cleavage from the other restriction enzymes. We appreciate the question and have now added this explanation to the revised manuscript.

One can't always tell which figure is being discussed when conclusions are drawn – a figure in the text versus a figure in supplementary information. For example, the statement about gap sizes does not follow from Figure S1 (p. 8, line 6 from bottom of page). The logic behind other statements is also not obvious, e.g., p. 14 – lines 4-5 from bottom of page, p. 16 – ¶ 2 lines 4-5, p. 25 – ¶2, lines 5-7.

RESPONSE: We thank the reviewer for pointing out where we failed to well match the text with the figures being discussed. We have carefully revised the manuscript to make sure it is clear what we are referring to.

It would be very helpful if all the sequence-related information about the minicircle (from the methods section and figures) were consolidated so that the reader could see the precise locations of enzyme cutting sites, nuclease recognition sites, attR curvature fragment, Nb.BbvCI site, etc. at the base-pair level. Such information might lead the reader to a better understanding of the logic used in the interpretation of data and appreciation of the information that has been gleaned from this research. An introductory image might also help the reader appreciate the problem under investigation and get a better understanding of the jargon used in describing the configurations of supercoiled DNA.

RESPONSE: We agree and now list the sequence (in Methods) with annotation to show precisely the location of the restriction sites and the attR site. We also now inform the reader that Nb.BbvCI nicks at the BbvCI site and the center of intrinsic curvature is centered around the MseI site.

The casual presentation is sometimes misleading. For example, the definition twist is imprecise (p. 4) – two pathways, not three as suggested in the text, define twist. Linking number is not a term (p. 4). Persistence-length DNA is not essentially rigid (p. 5) – rigid DNA is about an order of magnitude smaller than the persistence length (10's of bps such as the gold-nanoparticle labeled fragments studied by Per Harbury).

RESPONSE: We never intended three pathways as a definition of twist and are not quite sure how it was read in this way. We nonetheless reworded the definition and deleted the word “term” from the definition of linking number in the revised manuscript. We also removed “rigid” as a descriptor of a persistence length.

The cooperative effects observed experimentally by Kim et al. (Probing allostery through DNA. Science 2013, 339, 816-819) are relevant to the discussion on p. 6. The system, while linear, constrained by fixed ends.

RESPONSE: We reread this interesting paper and the reviewer is correct that the DNA used is linear but it is only constrained on one end. The allosteric effects observed are interesting but different from what we observe. Most notably, the effects reported in Kim et al. are relatively short-range and dampen within a few helical turns. The cooperative effects we observe are long-range mechanical correlations. To try to link it to our results required too much additional explanation, so we decided not to include it in the revised manuscript, although it is a worthy paper.

Figure 2B - the rotational alignment is not clear

RESPONSE: We assume the reviewer meant Figure 2A and agree that the two cylinders in our original version were not clear enough, so we have added a side-on view to better show the rotational angles in the revised figure.

Figure 6A – typo “stain”

RESPONSE: We thank the reviewer for catching this typo. It is corrected in the revised figure legend.

Figure 6B – not clear which loop is described as the smaller plectonemic loop

RESPONSE: We thank the reviewer for pointing this out. In the revised manuscript, we fixed the legend of Figure 6B to clarify that the loop containing the disrupted base would become smaller.

Figure S3 – Michaelis-Menten analysis unclear (symbols not defined)

RESPONSE: We appreciate the reviewer catching our omission and have now properly defined all the Michaelis-Menten analysis symbols in the legend of what was Supplementary Figure 3 in the original manuscript but is Supplementary Figure 2 in the revised manuscript.

REVIEWERS' COMMENTS

Reviewer #1 (Remarks to the Author):

I thank the authors for their response to my points and for including an additional schematic and table which I believe are both useful. I would recommend publication of this manuscript with no further changes.

Alice Pyne

Reviewer #2 (Remarks to the Author):

The authors have prepared a scholarly response that appropriately addressed all of the issues I raised and, I believe, have also addressed all of the issues of the other reviewers.

Reviewer #3 (Remarks to the Author):

The authors have responded very thoughtfully to the comments of this reviewer (and of other reviewers) on the first draft of their paper. The revised manuscript is substantially clearer than the original, making it easy for the reader to appreciate the structural insights into DNA supercoiling that they have gleaned from clever biochemical experiments.

I suggest accepting the paper after the authors address the following minor points:

1. The revised text (p. 4) is still somewhat misleading regarding twist, writhe, and linking number. Mathematically, writhe is based on one of the two curves used to define twist and linking number. The text suggests that three curves are involved. The description of writhe in terms of "the coiling of the DNA double helices about one another," may lead the reader to think that one of the curves is the pathway of the double-helical axis. If one chooses to describe twist and linking number in terms of the two DNA strands, the writhing number describes the pathway of one of the strands. In the case of canonical B DNA, the atoms on complementary strands, e.g., corresponding atoms of complementary residues on the two strands, describe a ribbon that resembles the ribbon described by the points along the DNA axis and one of the strands. The similarity of these two ribbons for B DNA may lead one to interpret DNA topology in terms of three curves.
2. The annotations of the restriction and attR sites on the 336-bp minicircle and the precise differences in the sequences of other constructs from this reference, now presented in the Methods section, are extremely helpful. What remains missing is explicit information on the base numbering scheme so that site locations reported in Supplementary Tables 2, 3 are crystal clear. That is, are the bases listed in the order 1-336? It would also be helpful if the authors explicitly described, in the text and figure legends, what appears to be a connection between the use of boldface in the description of the attR site, i.e., the boldface letters in the base sequence, and the thick arrows used in the schematics of the minicircle in Figures 3 and 5.
3. The legend to Figure 3 suggests that the curved fragment within the attR site is highlighted in Figure S1. Where is this sequence? As noted above, it would be helpful if the thick curved arrow denoting the attR site and the location of the start/end of the minicircle were clarified. The numerical values (presumably number of base pairs between the centers of successive cutting sites) also need some explanation.
4. The text (p. 22) suggests that site 2 is explicitly labeled in Figure 5c but the reader sees only the term Bal31 and does not know without further explanation if the tip of the arrow points to the center of site 2.
5. The term "bend the DNA towards the minor groove" used in the text (p. 26) and rebuttal (p. 2) is confusing. Does this mean that the minor groove is getting narrower, i.e., the width of the minor groove is less than in B DNA? DNA can kink in two directions – into both the major and minor

grooves. Whereas the DNA on the nucleosome bends sharply into the minor groove (via so-called negative roll), the DNA associated with numerous other proteins (e.g., DNA gyrase, RNA polymerase, Cre-loxP) bends equally if not more strongly into the major groove (via so-called positive roll). The deformations of DNA generated in atomic-level computer simulations also tend to narrow the major groove. Moreover, sharp bending into the major groove widens the minor groove and often occurs in concert with a large decrease local twist.

REVIEWERS' COMMENTS

Reviewer #1 (Remarks to the Author):

I thanks the authors for their response to my points and for including an additional schematic and table which I believe are both useful. I would recommend publication of this manuscript with no further changes.

Alice Pyne

We thank Dr. Pyne for her thoughtful and constructive feedback.

Reviewer #2 (Remarks to the Author):

The authors have prepared a scholarly response that appropriately addressed all of the issues I raised and, I believe have also addressed all of the issues of the other reviewers.

We thank reviewer 2 for their helpful and constructive comments.

Reviewer #3 (Remarks to the Author):

The authors have responded very thoughtfully to the comments of this reviewer (and of other reviewers) on the first draft of their paper. The revised manuscript is substantially clearer than the original, making it easy for the reader to appreciate the structural insights into DNA supercoiling that they have gleaned from clever biochemical experiments.

We thank reviewer 3 for their feedback on the previous draft which helped us to improve the readability of the paper. We address this reviewer's new comments below.

I suggest accepting the paper after the authors address the following minor points:

- 1. The revised text (p. 4) is still somewhat misleading regarding twist, writhe, and linking number. Mathematically, writhe is based on one of the two curves used to define twist and linking number. The text suggests that three curves are involved. The description of writhe in terms of "the coiling of the DNA double helices about one another," may lead the reader to think that one of the curves is the pathway of the double-helical axis. If one chooses to describe twist and linking number in terms of the two DNA strands, the writhing number describes the pathway of one of the strands. In the case of canonical B DNA, the atoms on complementary strands, e.g., corresponding atoms of complementary residues on the two strands, describe a ribbon that resembles the ribbon described by the points along the DNA axis and one of the strands. The similarity of these two ribbons for B DNA may lead one to interpret DNA topology in terms of three curves.*

We felt confident that our definitions of twist, writhe, and linking number were correct, even in the first submission but we did our best to understand where the reviewer had a problem. This time we consulted with Professor De Witt Sumners, a mathematician with expertise in DNA topology for a second opinion. Professor Sumners confirmed that the definitions we provided in the introduction are accurate. Writhe is used to measure how much the DNA molecule winds around itself, not how much the single DNA molecules wind around the axis of the molecule.

2. The annotations of the restriction and attR sites on the 336-bp minicircle and the precise differences in the sequences of other constructs from this reference, now presented in the Methods section, are extremely helpful. What remains missing is explicit information on the base numbering scheme so that site locations reported in Supplementary Tables 2, 3 are crystal clear. That is, are the bases listed in the order 1-336? It would also be helpful if the authors explicitly described, in the text and figure legends, what appears to be a connection between the use of boldface in the description of the attR site, i.e., the boldface letters in the base sequence, and the thick arrows used in the schematics of the minicircle in Figures 3 and 5.

Figure 3 has been further revised to show the designated start/end of the minicircle sequence and the sequence positions of the restriction sites on the map of the minicircle. Additionally, we added a sentence to the Methods section of the second revised manuscript to clarify that the bases are listed in the order 1–336. Boldface was used to highlight the *attR* sequence (without requiring the use of a different color). The thick arrow shows the location of the *attR* site, which is how we have done it in previous publications.

3. The legend to Figure 3 suggests that the curved fragment within the attR site is highlighted in Figure S1. Where is this sequence? As noted above, it would be helpful if the thick curved arrow denoting the attR site and the location of the start/end of the minicircle were clarified. The numerical values (presumably number of base pairs between the centers of successive cutting sites) also need some explanation.

The resolution of our comparative gel electrophoresis assay does not provide a precise location of the curved sequence but does show that it is centered around the MseI site as stated in the figure legend. We revised Supplementary Figure 1 to better highlight the location of intrinsic curvature. As described in our response above we have also revised Figure 3 to denote the designated start/end of the minicircle sequence. The meaning of the base pair distances listed on the schematic have also been clarified in the figure legend.

4. The text (p. 22) suggests that site 2 is explicitly labeled in Figure 5c but the reader sees only the term Bal31 and does not know without further explanation if the tip of the arrow points to the center of site 2.

Figure 5c has now been revised to show the locations of sites 1, 2, and 3 on the 672 bp minicircle. The arrow now points to the site 2 label.

5. The term “bend the DNA towards the minor groove” used in the text (p. 26) and rebuttal (p. 2) is confusing. Does this mean that the minor groove is getting narrower, i.e., the width of the minor groove is less than in B DNA? DNA can kink in two directions – into both the major and minor grooves. Whereas the DNA on the nucleosome bends sharply into the minor groove (via so-called negative roll), the DNA associated with numerous other proteins (e.g., DNA gyrase, RNA polymerase, Cre-loxP) bends equally if not more strongly into the major groove (via so-called positive roll). The deformations of DNA generated in atomic-level computer simulations also tend to narrow the major groove. Moreover, sharp bending into the major groove widens the minor groove and often occurs in concert with a large decrease local twist.

The term “bend the DNA towards the minor groove” is referring to the kink” observed in the computational simulations reported by Lankas et al. (2006). When we say “bend the DNA towards the minor groove” we are referring to the direction of the bend as it relates to the rotational register of the helix. The sentence has been revised to make this clearer.